# AhR diminishes the efficacy of chemotherapy via suppressing STING dependent type-I interferon in bladder cancer

Zikun Ma[1,2,9], Zhiyong Li[1,2,9], Yize Mao[1,3,9], Jingwei Ye[1,2,9], Zefu Liu[4,9], Yuzhao Wang[1,2], Chen Wei[5], Jun Cui [6] ✉, Zhuowei Liu [1,2,7] ✉ & Xiaoyu Liang [1,8] ✉

The induction of type-I interferons (IFN-Is) is important for the efficacy of chemotherapy. By investigating the role of amino acids in regulation of IFN-I production under chemo-drug treatment in bladder cancer (BC) cells, we find an inherent AhR-dependent negative feedback to restrain STING signaling and IFN-I production. Mechanistically, in a ligand dependent manner, AhR bridges STING and CUL4B/RBX1 E3 ligase complex, facilitating STING degradation through ubiquitin-proteasome pathway. Inhibition of AhR increases STING levels and reduces tumor growth under cisplatin or STING agonist treatment. Endogenous AhR ligands are mainly consisted of tryptophan (Trp) metabolites; dietary Trp restriction, blocking the key Trp metabolism rate-limiting enzyme IDO1 or inhibition of cellular Trp importation also show similar effect as AhR inhibition. Clinically, BC patients with higher intratumoral expression of AhR or stronger intratumoral Trp metabolism (higher IDO1 or Kyn levels) that lead to higher AhR activation show worse response rate to neoadjuvant chemotherapy (NAC).

Bladder cancer (BC) is one of the most common tumors and approximately 570,000 new cases are diagnosed yearly with 170,000 related deaths worldwide[1]. Depending on the absence or presence of muscle invasion, bladder cancer can be classified into non-muscle-invasive bladder cancer (NMIBC) and muscle-invasive bladder cancer (MIBC)[2,3]. The prognosis of NMIBC is generally favorable, whereas MIBC accounts for most related deaths[2]. The current primary treatment for MIBC is neoadjuvant chemotherapy (NAC) combined with radical cystectomy (RC)[3,4]. However, unlike other cancers where patients have benefited from the kinase-targeted therapy innovation,

the chemotherapy regimen has largely remained unchanged in MIBC for decades[4,5]. In fact, cisplatin-based regimens remain the mainstay of NAC for MIBC[4]. Therefore, based on reality, it is of great significance to improve the efficacy of the current NAC regimens for MIBC.

Previous studies indicated that the efficacy of chemotherapy was determined by T-cell-related immunity[6–8]. IFN-I signaling is essential for specific antitumor T-cell response[9,10]. More importantly, a variety of chemotherapeutic drugs, including cisplatin, induce the production of IFN-I[11,12]. The production of IFN-I mainly depends on the activation of pattern recognition receptors (PRRs) of the innate immune system[13,14].

[1]State Key Laboratory of Oncology in South China, Guangdong Key Laboratory of Nasopharyngeal Carcinoma Diagnosis and Therapy, Sun Yat-sen University Cancer Center, Guangzhou 510060, P. R. China. [2]Department of Urology, Sun Yat-sen University Cancer Center, Guangzhou 510060, P. R. China. [3]Department of Pancreatobiliary Surgery, Sun Yat-sen University Cancer Center, Guangzhou 510060, P. R. China. [4]Department of Urology, Xiangya Hospital, Central South University, Changsha 410008, P. R. China. [5]College of Life Sciences, University of Chinese Academy of Sciences, Beijing 100049, P. R. China. [6]MOE Key Laboratory of Gene Function and Regulation, State Key Laboratory of Biocontrol, School of Life Sciences, Sun Yat-sen University, Guangzhou 510275, P. R. China. [7]Sun Yat-sen University Cancer Center Gansu Hospital, Lanzhou 730050, P. R. China. [8]Department of Radiation Oncology, Sun Yat-sen University Cancer Center, Guangzhou 510060, P. R. China. [9]These authors contributed equally: Zikun Ma, Zhiyong Li, Yize Mao, Jingwei Ye, Zefu Liu. ✉e-mail: cuij5@mail.sysu.edu.cn; liuzhw@sysucc.org.cn; liangxy1@sysucc.org.cn

Among these, cGAS-STING (Cyclic GMP-AMP Synthase [cGAS], and Stimulator of Interferon Response CGAMP Interactor 1 [STING]) signaling plays a particularly critical role in tumors by recognizing free double-stranded DNA (dsDNA) in the cytoplasm[15,16]. It is also the core mechanism of various chemotherapeutic drugs, including cisplatin to mediate immune response[7,14,17]. However, this pathway appears to be absent or attenuated in multiple cancer types[18]. This observation, from another angle, suggests that tumor formation and progression require corresponding mechanisms to escape surveillance by cGAS-STING signaling[19]. Therefore, exploring the key molecules and mechanisms associated with this phenomenon is of great significance not only for improving chemotherapy-mediated immune response but also for understanding the process of tumor immune escape.

Dysregulation of tumor cell metabolism is involved in shaping an immunosuppressive tumor microenvironment[20]. Compared with metabolites of sugars and lipids, metabolites of amino acids, especially those of essential amino acids, are usually involved in signal transduction and have direct biological activities[21]. However, the key amino acids and mechanism in regulation of IFN-I production in context of chemotherapy is not well elucidated. In this work, by an unbiased screening, we find that Trp and its metabolites (L-kynurenine [Kyn], 5-hydroxytryptophan [5HTP], kynurenic acid [KynA], and tryptamine) actively dampen IFN-I production. Moreover, by multidisciplinary approach, we clarify the inner mechanism; explore the potential of Trp restriction dietary or pharmacologic inhibition of related molecules in sensitizing bladder cancer to chemo-drugs or STING agonist; and verify the clinical relevance with related genes, metabolic pathways, or metabolites.

## Results

### Trp diminishes IFN-I production induced by cisplatin

Previous studies indicated that the efficacy of chemotherapy was highly influenced by T-cell-related immunity[6–8]. To further confirm this in BC, we analyzed the relation between T-cell infiltration and response rate to NAC in patients of two public datasets[22,23]. As expected, higher expression of intratumoral cytotoxic T cells (CD3/CD8) was correlated with better response to NAC in patients of both datasets (Supplementary Fig. 1A, B). In Sjodahl et al's cohort, higher intratumoral T-cell levels also predicted better survival (Supplementary Fig. 1C, D). Furthermore, T-cell-based antitumor immunity was highly dependent on IFN-Is signaling[9]. In line with this, we found that the IFN-Is signaling and T-cell infiltration were highly associated in these two public datasets (Supplementary Fig. 1E−H), and patients with better NAC response also demonstrated increased expression of gene sets related to IFN-Is signaling (Supplementary Fig. 1I, J). In tumors, as revealed by previous studies, IFN-Is were mainly derived from tumor cells or myeloid cells autonomously and increased in the presence of external stimuli (radiation, chemotherapy)[24,25]. We analyzed the expression of all known 17 subtypes of human IFN-Is in tumor and myeloid cells isolated from fresh biopsies of patients with BC; and found that most of the subtypes that could be detected were at comparable levels between these two types of cells, and some were even expressed higher in tumor cells (Supplementary Fig. 1K). Moreover, in our recent study[26], we evaluated the ability to produce IFN-Is in tumor cells, fibroblasts, and myeloid cells through analyzing the expression levels of three different IFN-Is production-related data sets, and found that tumor cells universally showed the highest expression. More importantly, tumor cells were found to be the main component of the tumor mass in BC (Supplementary Fig. 1L); thus, above results suggested that they were the major source of IFN-Is.

It was reported that tumor cells exhibited dysregulation of amino acid metabolism[27,28]. To investigate whether this dysregulation interferes with IFN-Is production under chemotherapy, we transfected an in-house established primary BC cell line (SYBC1) with an *IFNB1* promoter reporter plasmid, and conducted an unbiased screening by

omitting specific individual essential or non-essential amino acids from the cell culture medium and treated cells with cisplatin, which was the backbone of NAC regimens for BC. As expected, cisplatin induced robust IFN-β expression (Fig. 1A). Notably, we found that cisplatin-treated cells cultured in medium without Trp showed enhanced IFN-β expression. In contrast, omitting threonine (Thr) largely decreased *IFNB1* expression, whereas cells cultured in medium lacking other amino acids did not show significant change on IFN-β expression (Fig. 1A). In line with this, by evaluating the consumption of amino acids, we found that cisplatin greatly enhanced the intake of Trp, methionine (Met), and glutamine (Glu), whereas the consumption of other amino acids was not affected (Fig. 1B).

Then, we conducted an enzyme-linked immunosorbent assay (ELISA) to detect the secreted IFN-β and quantitative polymerase chain reaction (qPCR) to test the expression of IFN-stimulated genes (ISGs) that participated in antigen presentation (major histocompatibility complex, class I, A (HLA-A)) or T-cell infiltration (C-X-C Motif Chemokine Ligand 10(CXCL10)). Consistently, in both SYBC1 and another BC cell line (UMUC-3), we found that omitting Trp or Thr showed similar effect as in forementioned screen assays (Fig. 1C, D and Supplementary Fig. 1M). Trp plays an immunosuppressive role, especially in regulating T-cell-related adaptive immunity, whereas its role in regulating IFN-Is production has not been well investigated. To further confirm this effect, we supplemented Trp in Trp omitted conditioned medium and evaluated IFN-Is production and consequent effect through tests including luciferase assay, ELISA, and qPCR; we found that with the increase of Trp concentration, the production of IFN-Is as well as expression of downstream ISGs further decreased (Fig. 1E, F and Supplementary Fig. 1N). Other stimuli, including HT-DNA, 3p-RNA and LPS can also induce cells to express IFN-Is, we transfected SYBC1 cells with HT-DNA, 3p-RNA or directly treated with LPS; then these cells were cultured in medium with or without Trp. After 24 h, we detected the expression of *IFNB1* via qPCR, and found that the omitting Trp had the most obvious effect on promoting *IFNB1* expression in scenario of HT-DNA transfection; removing Trp also enhanced the expression of *IFNB1* induced by 3p-RNA but the effect was much weaker compared with HT-DNA transfection; while, omitting Trp showed no effect on the effect of LPS (Supplementary Fig. 1O).

Next, we sought to verify the in vivo role of Trp in regulation IFN-Is production and related antitumor immunity under cisplatin treatment. Firstly, we confirmed the decrease of serum Trp levels in mice receiving Trp-free diet (Supplementary Fig. 1P) and no obvious side effects within the planned time. Then, we subcutaneously injected the murine bladder cancer cells MB49 into the flank of C57BL/6 mice to construct an ectopic implantation model. Tumor-bearing mice were fed with normal or Trp-free diet in the indicated manner, and we found that Trp restriction dietary reduced tumor growth under cisplatin treatment (Fig. 1G). Furthermore, in a parallel experiment, we intraperitoneally injected mice with CD8a antibody to eliminate murine CD8⁺ T cells and found that the chemo-sensitizing effect induced by restricting Trp intake was almost abolished (Fig. 1H). In agreement, after cisplatin treatment, both CD8⁺ T-cell infiltration and functional state were improved when Trp intake was limited (Fig. 1I). To further elucidate the effect of Trp on IFN-Is signaling, intratumoral levels of IFN-Is and their sequential downstream effector molecules (*H2-kb* and *Cxcl10*) were detected using qPCR. As expected, limiting Trp intake boosted the intratumoral levels of IFN-Is and downstream effector molecules (Supplementary Fig. 1Q).

Moreover, we also constructed an *Irf3*-knockout (KO) MB49 cell line using clustered regularly interspaced short palindromic repeats (CRISPR) technology (Supplementary Fig. 1R). This transgenic cell-line almost abolished IFN-Is production under cisplatin treatment (Fig. 1K). Mice bearing *Irf3*-KO MB49 cells were treated with the same dose of

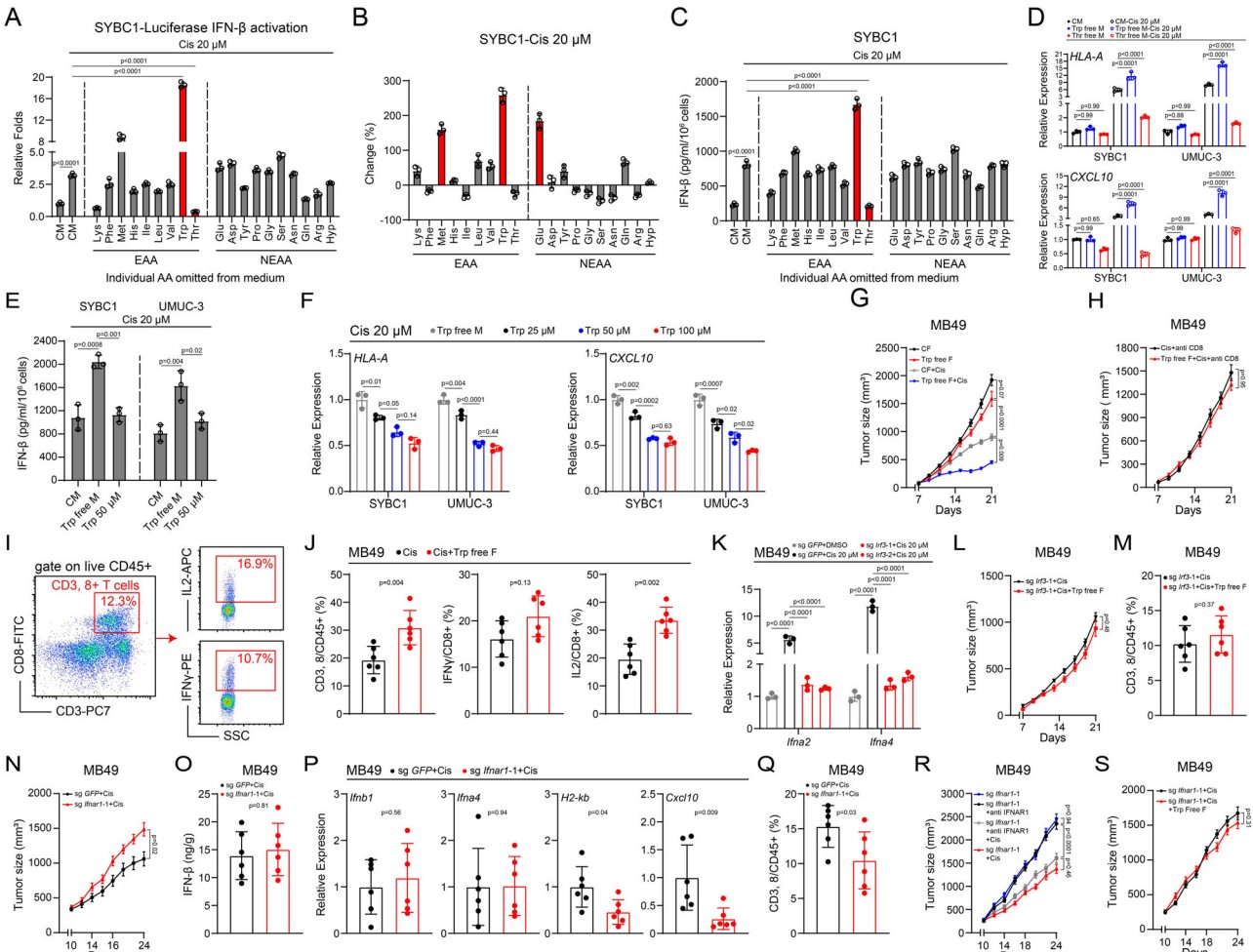

**Fig. 1 | Trp diminishes IFN-I production induced by cisplatin. A** Luciferase reporter assays for IFN-β after individual amino acid (AA) omitted from medium and treated with cisplatin. **B** HPLC-MS detection of AA consumption by cultured tumor cells (bars show change from fresh medium). **C** ELISA for IFN-β content in the supernatant after individual AA omitted from medium and treated with cisplatin. **D** mRNA expression of *HLA-A* and *CXCL10* after indicated treatment. **E** ELISA for IFN-β content in the supernatant after Trp-free medium culture or Trp supplementation and treated with cisplatin. **F** mRNA expression of *HLA-A* and *CXCL10* after Trp-free medium culture and treated with cisplatin. **G** Effect of Trp-free feed combined with cisplatin on MB49 growth (*n* = 6) (Mean ± SEM). **H** Effect of Trp-free feed combined cisplatin and anti-CD8 treatment on MB49 growth (*n* = 6) (Mean ± SEM). **I, J** Flow cytometry gating strategy (**I**) and effect of Trp-free feed combined with cisplatin on percentage of CD8⁺ T cells, IFNγ⁺CD8⁺ T cells, and IL2⁺CD8⁺ T cells in MB49-bearing mice (*n* = 6) (**J**). **K** mRNA expression of *Ifna2* and *Ifna4* in *Irf3*

knocked-out cells after treated with cisplatin. **L, M** Effect of *Irf3* knocked-out combined cisplatin and Trp-free feed on MB49 growth (*n* = 6) (Mean ± SEM) (**L**), and percentage of CD8⁺ T cells (**M**) in tumor. **N–Q** Effect of *Ifnar1* knocked-out combined with cisplatin on MB49 growth (*n* = 6) (Mean ± SEM) (**N**), IFN-β content in tumor (**O**), mRNA expression of *Ifnb1*, *Ifna4*, *H2-kb*, and *Cxcl10* (P), and percentage of CD8⁺ T cells (**Q**) in tumor. **R** Effect of *Ifnar1* knocked-out combined with anti-IFNAR1 neutralizing antibody and cisplatin on MB49 growth (*n* = 6) (Mean ± SEM). **S** Effect of *Ifnar1* knocked-out combined with Trp-free feed and cisplatin on MB49 growth (*n* = 6) (Mean ± SEM). *P* value by One-way ANOVA (**A, C–G, K, R**). *P* value by two-tailed Wilcoxon (**H, J, L–Q, S**). All *p* value < 0.05 as statistic difference. Error bars represent Mean ± SD, unless otherwise indicated. Three biologically independent experiments were performed (**A–F, K**). Source data are provided as a Source Data file.

cisplatin used in the previous experiments and accepted Trp restriction dietary. Under these conditions, we found that the volumes of *Irf3*-KO MB49-formed tumors and the numbers of intratumoral CD8⁺ T cells were not affected by Trp restriction (Fig. 1L, M). On the other hand, we also constructed an *Ifnar1*-KO MB49 cell line and found that sg *Ifnar1*-MB49 cells formed larger tumors than sg *GFP*-MB49 cells (Supplementary Fig. 1S, T). After cisplatin treatment, tumors formed by sg *Ifnar1*-MB49 cells were also larger, while the difference was bigger than the scenario without cisplatin treatment (Fig. 1N). Furthermore, we collected tumors after the first dose of cisplatin and detected the content of intratumoral IFN-β, the expression of key ISGs (*H2-kb*, *Cxcl10*), IFN-Is (*Ifnb1*, *Ifna4*) and the changes of intratumoral T cells. We found that after knocking-out *Ifnar1*, there was no significant change in intratumoral IFN-β levels and the expression of *Ifnb1* and *Ifna4*; while the expression of these key ISGs (*H2-kb*, *Cxcl10*) and

the T-cell infiltration were decreased (Fig. 1O–Q). Then, we further explored whether IFN-Is affected the efficacy of cisplatin through other non-tumor cells by using IFNAR1 blocking antibody. In the absence of cisplatin treatment, sg *Ifnar1*-MB49 cells formed tumors of almost the same sizes regardless of injection of IFNAR1 neutralizing antibody (Fig. 1R). Under cisplatin treatment, intraperitoneal injection of IFNAR1 neutralizing antibody led to larger but not statistically different tumor volumes (Fig. 1R). In line with this, there were no differences in the expression of key ISGs in the presence of IFNAR1 neutralizing antibody, but the T-cell infiltration showed decreased trend (Supplementary Fig. 1U). More importantly, Trp-free diet did not enhance the effect of cisplatin in sg *Ifnar1*-MB49 tumors (Fig. 1S). Taken together, these in vitro and in vivo data suggested that Trp diminished IFN-Is production induced by cisplatin which led to dampened T-cell dependent antitumor immunity.

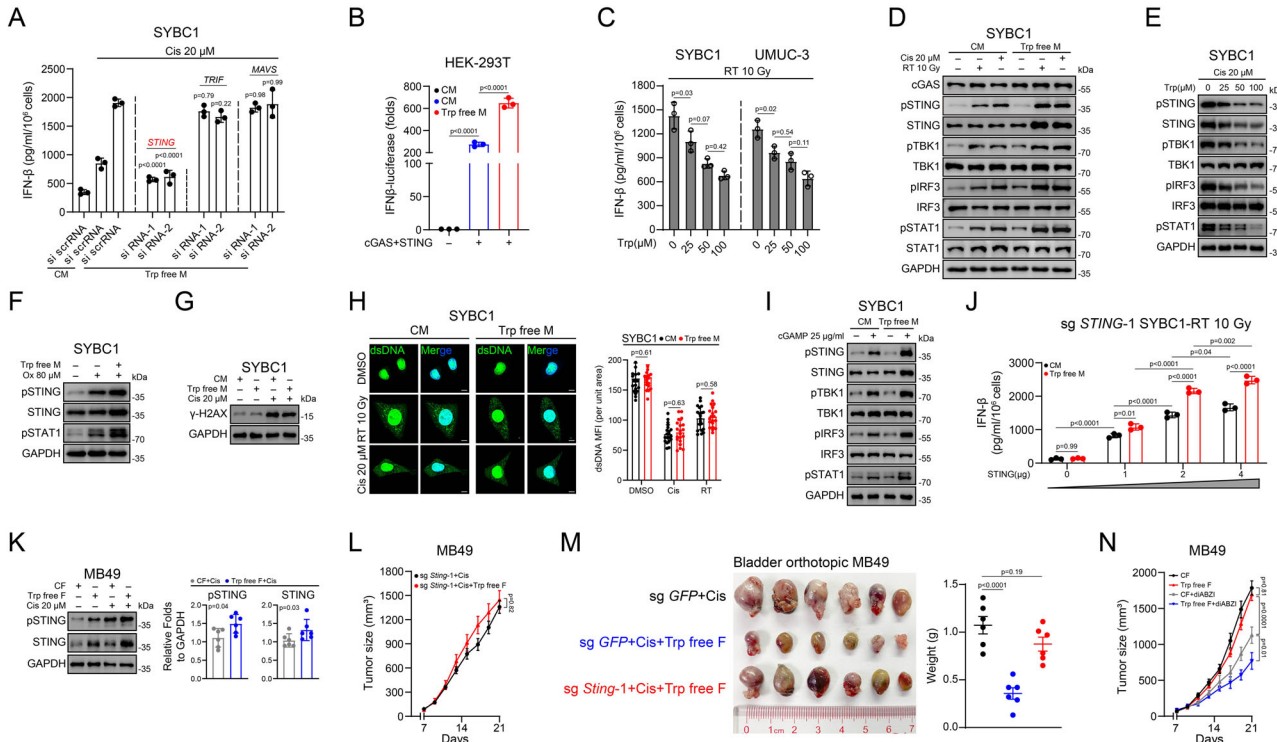

**Fig. 2 | Trp suppresses IFN-I production at STING level. A** ELISA for IFN-β content in the supernatant after knocked-down indicated genes and cultured in CM or Trp-free medium. *P* value compared to si-scrRNA in Trp-free medium group. **B** Luciferase reporter assays for IFN-β after transfected with luciferase reporters for cGAS alone with STING. **C** ELISA for IFN-β content in the supernatant after Trp supplementation and treated with RT. **D** Immunoassay for representative cGAS-STING related genes expression after treated with cisplatin or RT under indicated medium. **E** Immunoassay for representative cGAS-STING related genes expression after Trp supplementation and treated with cisplatin. **F** Immunoassay for representative cGAS-STING related genes expression after treated with oxaliplatin under indicated medium. **G** Immunoassay for H2A.X expression after cultured in CM or Trp-free medium and treated with cisplatin. **H** Representative immunofluorescent staining images for dsDNA expression and location after treated with cisplatin or RT under indicated medium (*n* = 20 cells per group). Scale bar 20 μm. **I** Immunoassay for cGAS-STING-related genes expression after treated with cGAMP under indicated medium. **J** ELISA for IFN-β content in the supernatant after transfected with STING under the indicated medium. **K** Immunoassay of pSTING and STING expression in MB49-bearing mice after indicated treatment (**n** = 6). **L** Effect of *Sting* knocked-out combined cisplatin and Trp-free feed on MB49 growth (*n* = 6) (Mean ± SEM). **M** Effect of *Sting* knocked-out combined cisplatin and Trp-free feed on bladder orthotopic MB49 growth (*n* = 6). **N** Effect of Trp-free feed combined diABZI on MB49 growth (*n* = 6) (Mean ± SEM). *P* value by One-way ANOVA (**A–C, J, M, N**). *P* value by two-tailed Wilcoxon (**H, K, L**). All *p* value < 0.05 as statistic difference. Error bars represent Mean ± SD, unless otherwise indicated. Three biologically independent experiments were performed (**A–G, I, J**). Source data are provided as a Source Data file.

## Trp suppresses IFN-Is production at STING level

Adaptors of three different systems (STING, mitochondrial antiviral signaling protein [MAVS], and TIR-domain-containing adaptor-inducing interferon-β [TRIF])[29], which integrate stress and trigger IFN-Is production were screened using short interfering RNA (siRNA) (Supplementary Fig. 2A). We found that only knocking-down of *STING* almost abolished the induction of IFN-β by cisplatin in condition with or without Trp, whereas interfering with the other two adaptors showed no obvious effect (Fig. 2A and Supplementary Fig. 2B). To further confirm the involvement of cGAS-STING signaling, we co-transfected plasmids encoding cGAS, STING, and IFN-β luciferase reporter into HEK293T cells. The results showed that exogenous *cGAS* and *STING* expression autonomously induced *IFNB1* expression and was enhanced when Trp was omitted (Fig. 2B). In clinical practice, radiotherapy (RT) had also been adopted for BC treatment, which also intensely activated cGAS-STING signaling[14]. Consequently, we examined the regulatory influence of Trp on IFN-Is induction by RT. Being similar with cisplatin, omitting Trp increased the secretion of IFN-β and the expression of *HLA-A* and *CXCL10*, whereas in the rescue assays, the trends were reversed (Fig. 2C and Supplementary Fig. 2C). Furthermore, we also extended our investigation to HeLa and MCF-7 cells that derived from other tissues (Supplementary Fig. 2D).

After confirming the role of Trp in regulating cGAS-STING signaling phenotypically, we sought to elucidate the underlying mechanism. Consistent with previous studies, the basal phosphorylation levels of critical molecules of cGAS-STING (STING, TBK1, IRF3) and IFN-Is (STAT1) signaling were extremely low, whereas cisplatin or RT induced marked activation (Fig. 2D). In concordance with our previous results, omitting Trp further enhanced phosphorylation of STING, TBK1, IRF3, and STAT1 (Fig. 2D), while rescuing Trp reversed this trend (Fig. 2E and Supplementary Fig. 2E). Moreover, deprivation of Trp also exerted similar effect on STING and IFN-Is signaling under oxaliplatin treatment, another highly immunogenic platinum drug (Fig. 2F).

Notably, we observed that omitting Trp increased the protein level of STING but not that of other molecules (Fig. 2D). Thus, we speculated that the enhanced transcription of IFN-Is was due to upregulation of STING protein level and subsequent enhancement of downstream signaling. To verify this, we firstly excluded the possibility that omitting Trp enhanced double DNA breaks. Immunoblot analysis and Picogreen staining showed that DNA breakage levels and cytosolic dsDNA in cells cultured in complete medium were similar with those of cells cultured in medium without Trp (Fig. 2G, H and Supplementary Fig. 2F). Then, we measured the level of cGAMP, and found that the lack of Trp also does not affect the production of cGAMP (Supplementary Fig. 2G). However, in another test, it was found that the lack of Trp could enhance the effect of cGAMP (Fig. 2I). These results suggested that enhanced STING activation by Trp was not due to increased dsDNA breaks nor enhanced cGAS function. To exclude the

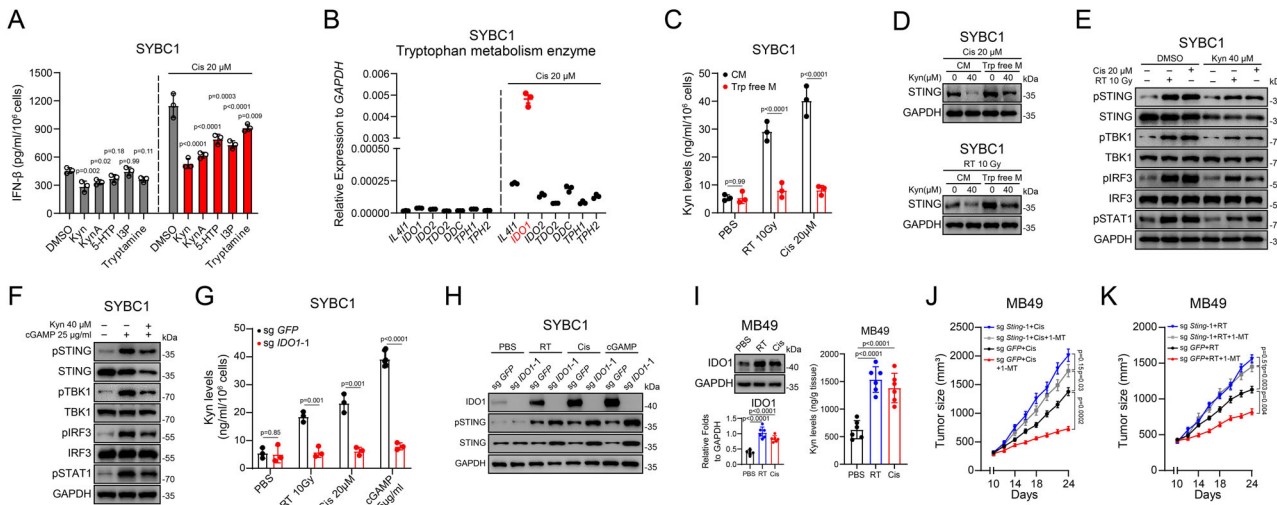

**Fig. 3 | Indoleamine 2,3-dioxygenase 1 (IDO1)-mediated Trp decreased STING level and IFN-I production. A** ELISA for IFN-β content in the supernatant after cisplatin combined indicated treatment. **B** mRNA expression of Trp catabolic enzymes after treated with cisplatin. **C** HPLC-MS detection of Kyn levels after treated with cisplatin or RT under indicated medium. **D** Immunoassay for STING expression after treated with Kyn under indicated medium. **E** Immunoassay for representative cGAS-STING related genes expression after treated with Kyn under indicated medium. **F** Immunoassay for representative cGAS-STING related genes expression after treated with cGAMP or Kyn. **G** HPLC-MS detection of Kyn levels for knocked-out *IDO1* cells after indicated treatment. **H** Immunoassay for IDO1, pSTING, and STING expression of *IDO1* knocked-out cells after indicated treatment. **I** Immunoassay for IDO1 expression (left) and HPLC-MS detection of Kyn levels (right) in MB49-bearing mice after treated with cisplatin or RT (**n** = 6). **J, K** Effect of cisplatin (J), and RT (K) combined with 1-MT on MB49 growth (n = 6) (Mean ± SEM). *P* value by One-way ANOVA (**A, I−K**). *P* value by two-tailed *t* test (**C, G**). All *p* value < 0.05 as statistic difference. Error bars represent Mean ± SD, unless otherwise indicated. Three biologically independent experiments were performed (**A−H**). Source data are provided as a Source Data file.

involvement of TBK1, we used 3p-RNA, which activated TBK1 via the RIG-I-MAVS pathway and found that omitting Trp did not significantly affect TBK1 activation (Supplementary Fig. 2H). Furthermore, we constructed a stable *STING*-KO cell line and re-overexpressed *STING*. Expectedly, *STING* re-overexpression rescued the effect of omitting Trp, and the extent of the effect was positively correlated with the expression level of STING (Fig. 2J and Supplementary Fig. 2I).

Next, we sought to verify the in vivo regulatory effect of Trp on cGAS-STING signaling. To this end, we detected the phosphorylation and expression levels of STING in tumor lysates from mice accepted Trp restriction dietary combined with cisplatin or RT. As expected, the phosphorylation and expression levels of STING were upregulated in mice fed with Trp-free diet (Fig. 2K and Supplementary Fig. 2J, K). Furthermore, knocking-out *Sting* abolished the in vivo effect led by Trp restriction not only in the subcutaneous tumor-bearing mouse model of MB49 cells but also in the orthotopic tumor model of MB49 cells (Fig. 2L, M and Supplementary Fig. 2L). We also tested the efficacy of limiting Trp intake on STING agonists and found Trp restriction dietary greatly reduced tumor growth under diABZI treatment (Fig. 2N). Taken together, our results indicated that Trp suppresses IFN-Is production through downregulation STING.

**Indoleamine 2,3-dioxygenase 1 (IDO1)-mediated Trp decreased STING level and IFN-Is production**
Next, we sought to clarify the mechanism by which Trp regulated the protein level of STING. We screened numerous endogenous Trp-derived metabolites and identified several (Kyn, KynA, 5HTP, indole-3-propionic acid [I3P], and tryptamine) that could decrease *IFNB1* expression (Fig. 3A)[30]. Furthermore, this also was accompanied by downregulation of STING protein level (Supplementary Fig. 3A). Currently, seven Trp catabolizing enzymes have been identified[31], which generated different metabolites including the ones we found with positive results in our screen (Supplementary Fig. 3B).

We analyzed the expression of the enzymes and found that all basal levels were extremely low, and only *IDO1* was dramatically induced after cisplatin treatment or RT (Fig. 3B and Supplementary

Fig. 3C, D). IDO1 catalyzes the conversion of Trp to Kyn. Therefore, we detected the intracellular Kyn levels using high-performance liquid chromatography-mass spectrometry (HPLC-MS), and found that both cisplatin and RT induced robust intracellular Kyn production in cells cultured in complete medium, whereas Kyn production was diminished in the Trp-free medium (Fig. 3C and Supplementary Fig. 3E).

To further verify the crucial role of the IDO1-Kyn metabolic pathway, we firstly examined the biological effect of Kyn on STING expression and downstream IFN-Is production-related molecules. It was found that in cells cultured with or without Trp under the treatment of cisplatin or RT, Kyn could decrease STING expression (Fig. 3D). Decreased expression of STING induced by Kyn diminished its phosphorylation and the phosphorylation of downstream signaling molecules (TBK1 and IRF3); consistently, IFN-β production was decreased as well (Fig. 3E and Supplementary Fig. 3F). Furthermore, we found that Kyn also diminished the effect of cGAMP (Fig. 3F) and verified a similar role of Kyn in human (MCF-7 and HeLa) and murine (MB49) tumor cell lines (Supplementary Fig. 3G).

Then, we constructed *IDO1*-KO SYBC1 cell line using CRISPR/Cas9 technology (Supplementary Fig. 3H), and Kyn production was abolished in the transgenic cells (Fig. 3G). The protein levels and phosphorylation of STING were enhanced in *IDO1*-KO cells cultured in complete medium under cisplatin, RT and cGAMP treatment (Fig. 3H); however, in cells without additional treatment, *IDO1* knocking-out showed minimal effect upon STING (Fig. 3H). Moreover, similar phenomenon was also observed when IDO1 inhibitor 1-methyl-tryptophan (1-MT) was added in the presence of oxaliplatin treatment (Supplementary Fig. 3I). Previous studies have shown that IDO1 could be induced by IFNs. STAT1 is the master transcription factor of IFN signaling;[32] therefore, we constructed a *STAT1*-KO cell line and found that IDO1 expression was not induced by RT, cisplatin, or cGAMP, whereas STING protein abundance and phosphorylation level were enhanced (Supplementary Fig. 3J, K). Then, we exogenously expressed IDO1 in STAT1-KO cell lines and found that IDO1 overexpression decreased STING protein abundance and activation (Supplementary Fig. 3L).

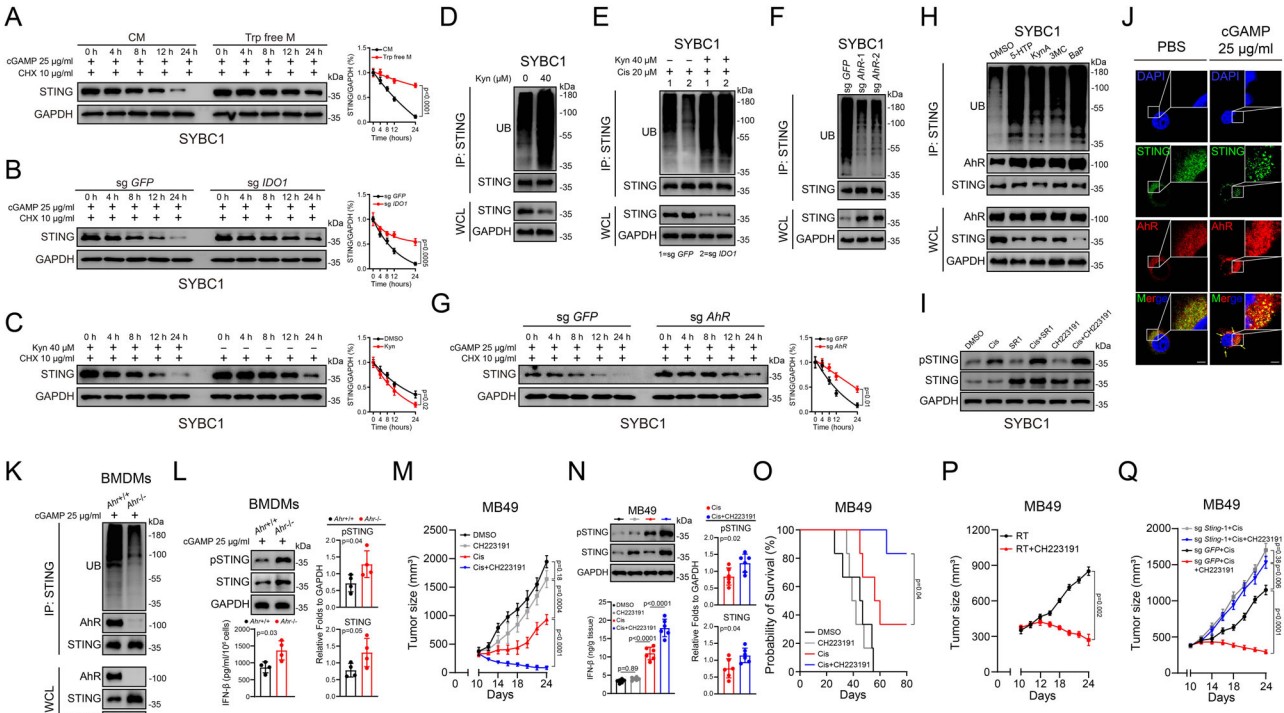

**Fig. 4 | Trp-IDO1-Kyn metabolic pathway regulated STING stability through AhR in a ubiquitin-proteasome-dependent manner. A–C** Immunoassay for STING expression after Trp-free M cultured (**A**), knocked-out *IDO1* (**B**), or treated with Kyn (**C**) in different time points with indicated treatment. **D** Coimmunoprecipitation and immunoassay for extracts of cell lysate by anti-STING antibodies after treated with Kyn. **E** Coimmunoprecipitation and immunoassay for extracts of cell lysate of *IDO1* knocked-out cells by anti-STING antibodies after treated with Kyn and cisplatin. **F** Coimmunoprecipitation and immunoassay for extracts of cell lysate of *AhR* knocked-out cells by anti-STING antibodies. **G** Immunoassay for STING expression after knocked-out *AhR* and treated with cGAMP at different time points. **H** Coimmunoprecipitation and immunoassay for extracts of cell lysate by anti-STING antibodies after indicated AhR agonist treatment. **I** Immunoassay for pSTING and STING expression after treated with SR1 and CH223191 combined with cisplatin. **J** Confocal microscopy for STING and AhR expression after treated with cGAMP. Scale bar 20 μm.

**K** Coimmunoprecipitation and immunoassay for extracts of cell lysate of BMDMs from *Lyzm-cre-AhR^fl/fl* mice after treated with cGAMP (*n* = 4). **L** Immunoassay of pSTING/ STING expression and IFN-β content in BMDMs isolated from *Lyzm-cre-AhR^fl/fl* mice (*n* = 4). **M, N** Effect of CH223191 combined with cisplatin on MB49 growth (*n* = 6) (mean ± SEM) (**M**), pSTING and STING expression, and IFN-β production (**N**) in tumor. **O** Kaplan–Meier survival curves of MB49-bearing mice after treated with CH223191 combine with cisplatin (*n* = 6). **P** Effect of CH223191 combined with RT on MB49 growth (*n* = 6) (mean ± SEM). **Q** Effect of cisplatin combined with CH223191 on MB49 growth (*n* = 6) (Mean ± SEM). *P* value by two-tailed *t* test (**A–C, G**). *P* value by two-tailed Wilcoxon (**L, N, P**). *P* value by one-way ANOVA (**M, N, Q**). *P* value by Kaplan–Meier survival analysis (**O**). All *p* value < 0.05 as statistic difference. Error bars represent Mean ± SD, unless otherwise indicated. Three biologically independent experiments were performed (**A–J**). Source data are provided as a Source Data file.

Furthermore, we sought to clarify the in vivo effect of the IDO1-Kyn metabolic pathway. In tumor lysates from mice received cisplatin treatment or RT, we found that Kyn and IDO1 levels were increased (Fig. 3I). Consistently, intratumorly injection of Kyn decreased expression and activation of STING as well as IFN-β production following cisplatin treatment and RT (Supplementary Fig. 3M). Moreover, pharmacologic inhibition of IDO1 with 1-MT not only reduced tumor growth but also increased STING level and activation under cisplatin, RT, or diABZI treatment (Fig. 3J, K and Supplementary Fig. 3N, O). Knocking-out *Sting* almost abolished the synergistic effect of IDO1 inhibitor with cisplatin or RT; meanwhile, 1-MT also promoted the infiltration and function of CD8+ T cells under cisplatin treatment (Supplementary Fig. 3P).

### Trp-IDO1-Kyn metabolic pathway regulated STING stability through AhR in a ubiquitin-proteasome-dependent manner

Next, we sought to uncover the mechanism by which the Trp-IDO1-Kyn metabolic pathway regulates STING expression. We found that neither Trp deprivation nor Kyn treatment affected the mRNA levels of *STING* in cells with or without Cisplatin treatment; moreover, mRNA levels of *STING* were also barely affected by knocking-out *IDO1* (Supplementary Fig. 4A). Thus, we speculated that changes in protein levels of STING was due to altered protein stability.

Thus, we used cycloheximide (CHX) to inhibit protein synthesis and found Trp deprivation greatly increased the half-life of STING under cGAMP or cisplatin treatment (Fig. 4A and Supplementary Fig. 4B); consistently, knocking-out *IDO1* also showed similar effects (Fig. 4B). Protein can be degraded via lysosomes or proteasomes; for STING, activated STING is mainly degraded through lysosomes[33–35]. However, at the cellular level, inevitably, both activated and unactivated STING co-existed in the cells, and the ratio of the two is inconsistent due to different stimuli. In our scenario, we found that Kyn could lead to a shortened half-life of STING even in the absence of its activation (Fig. 4C). Without activation, the protein levels of STING were also decreased in presence of Kyn and the downregulation could be reversed by MG132 (proteasome inhibitor) but not CQ (lysosome inhibitor) (Supplementary Fig. 4C). Therefore, we speculated that IDO1-Kyn metabolic pathway might selectively degrade of unactivated STING via proteasomes. Proteasomes degrade ubiquitinated proteins. As expected, by immunoprecipitation and immunoblotting, we found that Kyn treatment enhanced the ubiquitination of STING (Fig. 4D). In addition, knocking-out of *IDO1* in SYBC1 cells resulted in decreased ubiquitination of STING under the treatment of cisplatin or cGAMP; however, the downregulation of STING ubiquitination led by knocking-out of IDO1 was reversed in presence of Kyn. Consistently, deprivation of Trp in the medium also resulted in decreased levels of STING

ubiquitination, and addition of Kyn could also reverse this effect (Fig. 4E and Supplementary Fig. 4D, E). Moreover, in contrast to conditions in previous studies[33,34], STING activation was weaker under our conditions, and it did not show obvious decrease in protein levels after activation; these data also indicated that the SYBC1 cells had a weak ability to uptake cGAMP, and could not strongly activate STING without membrane permeabilization. Meanwhile, compared to CQ, MG132 was more capable on upregulating STING protein levels in our scenarios (Supplementary Fig. 4F). However, in the condition of strong activation of STING led by cGAMP treatment under membrane permeabilization, the degradation effect on STING of lysosome was more significant than that of proteasome (Supplementary Fig. 4G). These results suggested that proteasome-dependent STING degradation might be more important than lysosome in the case of weak STING activation; and activated as well as unactivated STING might follow different degradation modes.

Next, we sought to identify the key molecules involved in the established ubiquitin-proteasome-dependent regulation of STING by Trp-IDO1-Kyn axis. To examine the previously reported involvement of AhR in mediating the effects of Kyn[36,37], we inhibited AhR using two specific inhibitors, CH223191 and StemRegenin1 (SR1). Our results showed that inhibiting AhR not only abolished the effect of Kyn on STING stability, but it also decreased its ubiquitination level (Supplementary Fig. 4H). Moreover, similar results were observed in *AhR*-KO cells, which had a prolongation of the half-life of STING protein (Fig. 4F, G and Supplementary Fig. 4I). To test whether the effect of AhR was dependent on its transcriptional function, we treated tumor cells with actinomycin D, an mRNA synthesis inhibitor, and found that the effect of Kyn on *STING* expression was not influenced by inhibition of mRNA synthesis (Supplementary Fig. 4J). Thus, we speculated that AhR might directly interact with STING to affect its stability.

We next transfected plasmids expressing MYC-tagged STING or HA-tagged AhR individually or together into HEK293T cells and conducted coimmunoprecipitation assays. As expected, the exogenously expressed MYC-STING and HA-AhR could interact with each other, and the interaction could be enhanced by Kyn or abolished by SR1; consistently, STING ubiquitination levels varied with the changes of the interaction between AhR and STING (Supplementary Fig. 4K). Endogenous interaction between STING and AhR was also confirmed, and the interaction could be enhanced by not only Kyn but also other endogenous (5HTP and KynA) and exogenous (3MC and BaP) ligands of AhR[38]; and the ubiquitination levels of STING increased as the stronger interaction and decreased as the weaker interaction (Fig. 4H). In addition, we also noticed that inhibition of AhR increased STING protein level even in cells without external stimuli (Fig. 4I and Supplementary Fig. 4L). This observation suggested that AhR was kept activated at basal levels in tumor cells, and STING expression was continuously suppressed. Moreover, immunofluorescence assays showed that STING and AhR were colocalized, and this effect was enhanced by cGAMP (Fig. 4J). Consistently, when AhR inhibitor CH223191 was added in the presence of oxaliplatin treatment, not only protein levels of STING but also its phosphorylation was increased (Supplementary Fig. 4M). In line with previous results, bone marrow-derived macrophages (BMDMs) from *Lyzm-cre-AhR^{fl/fl}* mice showed enhanced STING stability and decreased ubiquitination under cGAMP treatment (Fig. 4K). After cGAMP treatment, *AhR*-KO BMDMs exhibited enhanced STING activation and IFN-β secretion (Fig. 4L). This observation suggested that the effect of AhR on STING might be highly conservative.

In in vivo experiments, we found that using AhR inhibitor CH223191 alone showed minimal effect upon limiting tumor growth while using CH223191 could largely reduce tumor growth under cisplatin, RT, or diABZI treatment; consistently, we also detected that CH223191 increased STING protein level and activation in tumor lysates as well as boosted intratumoral IFN-β levels (Fig. 4M-P and Supplementary Fig. 4N–Q). Moreover, in line with previous results,

knocking-out *Sting* almost abolished the synergistic effect of AhR inhibitor with cisplatin or RT (Fig. 4Q and Supplementary Fig. 4R); meanwhile, CH223191 also promoted the infiltration and function of CD8+ T cells under cisplatin treatment (Supplementary Fig. 4S).

## STING was ubiquitinated on lysine 236 with K48 linkage by AhR

Structurally, STING consists of two parts, the N-terminal in the transmembrane domain that spans the endoplasmic reticulum (ER) membrane and the C-terminal that faces the cytosol[15]. Based on this knowledge, we generated MYC-tagged STING truncated mutants by separating STING into its N (1–139 amino acids)- and C (140–379 amino acids)-terminals. Experiments with these mutants showed that the C-terminal mediated the interaction with AhR (Fig. 5A).

Moreover, AhR as a bHLH-PAS transcription factors, shares a typical structure with family members. Thus, we constructed truncated mutants by separating AhR into the MYC-tagged bHLH domain that binds DNA (1–120 amino acids), the PAS domain consisting of tandemly connected part A and B that mediated protein interaction or ligand binding (120–424 amino acids), the transcription activation domain (425–848 amino acids) and N-terminal that excluded the transcription activation domain (1–424 amino acids)[39]. We found that the transcription activation domain dominantly mediated the interaction with STING, while other domains barely interacted with STING (Fig. 5B); It was worth mentioning that previous studies reported that AhR mainly interacted with other proteins through the PAS domain[40–42]. Furthermore, AhR mutants that deleting PAS-B (277–424 amino acids) was in a constitutively activated state, showed enhanced interaction with STING (Fig. 5C).

To further clarify the details of the interaction between AhR and STING and the alteration of this interaction by endogenous AhR ligand Kyn, we performed molecular docking and molecular dynamics (MD) simulations. Normally, STING exists as homodimer on ER, thus we simulated the interaction between AhR and STING monomer and AhR and STING dimer respectively. As expected, through the analysis of root-mean square-deviation (RMSD), it was found that both the AhR-STING (monomer) and AhR-STING (dimer) complexes can form an equilibrium state, and this equilibrium state was easier to form in the presence of Kyn (Fig. 5D). Moreover, root-mean-square-fluctuation (RMSF) analysis indicated that after binding Kyn, the fluctuation of STING was more stable in both models; while AhR (chain A) was more stable in the complex interacting with the STING dimer than with the STING monomer (Supplementary Fig. 5A–C). Then, based on Molecular Mechanics Poisson-Boltzmann Surface Area (MMPBSA), we found that the bind free energy of single AhR molecule with single STING molecule in AhR-STING (monomer) and AhR-STING (dimer) were shifted from −127.36 kcal/mol to −171.43 kcal/mol and −142.13 kcal/mol to −214.66 kcal/mol respectively (Supplementary Table 1). Furthermore, we conducted more detailed analysis. In the equilibrium state of these complexes, the contact residues between AhR and STING in AhR-STING (monomer) and AhR-STING (dimer) were highly overlapping and most of the overlapping residues remained unchanged in the scenario of AhR binding with Kyn (Supplementary Tables 2–5). Consistent with the previous immunoprecipitation results, the residues that AhR interacted with STING were basically concentrated after the 600th amino acid, especially after AhR binding with Kyn (Supplementary Tables 2–5). Therefore, we speculated that in the real world, the interaction mode of two molecules of AhR and STING dimer in mirror-symmetry pattern was more likely to exist. Furthermore, the surface binding mode of AhR-STING and AhR-STING-KYN were shown. The end of the residues in AhR is extended to STING, acting like a 'crab claw' chelating with STING (Fig. 5E–H and Supplementary Movies 1–2), which resulted in stronger binding between AhR protein and STING protein. Importantly, above phenomenon was also confirmed in AhR-STING-KYN (Dimer) complex (Fig. 5I–L and Supplementary Movies 3–4).

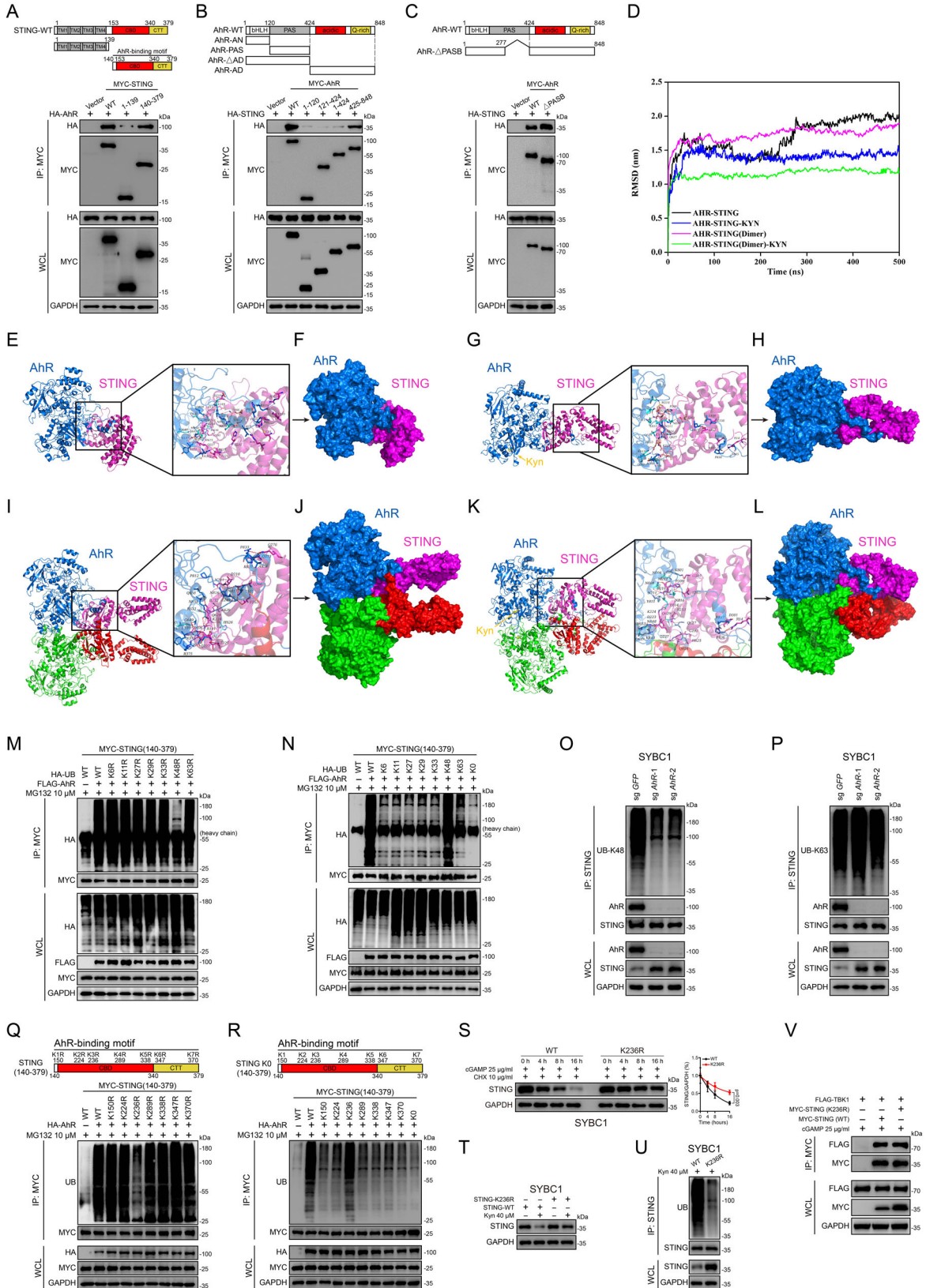

STING can be ubiquitinated at multiple sites and the forms of polyubiquitin chains linked are also variable, which confers different biological functions[15]. Investigations of a panel of ubiquitin mutants to elucidate the forms of polyubiquitin chains mediated by AhR showed that conversion of lysine to arginine at the K48 site markedly decreased polyubiquitinated STING (Fig. 5M and

Supplementary Fig. 5D). In addition, the K0 mutant ubiquitin showed that the ubiquitination reappeared when K48, rather than other lysine molecules, was reintroduced (Fig. 5N and Supplementary Fig. 5E). Furthermore, experiments with antibodies specifically targeting the K48 or K63 ubiquitin linkage showed that endogenous or exogenous AhR conferred K48 specific linkage

**Fig. 5 | STING was ubiquitinated on lysine 236 with K48 linkage by AhR.**
**A** Coimmunoprecipitation and immunoassay for extracts of HEK293T cells transfected with HA-AhR (WT), MYC-STING (WT), STING (1-139), and STING (140–379) by anti-MYC beads. **B** Coimmunoprecipitation and immunoassay for extracts of HEK293T cells transfected with MYC-AhR (WT), AhR (1-120), AhR (120–424), AhR (1–424), AhR (424–848), and HA-STING (WT) by anti-MYC beads.
**C** Coimmunoprecipitation and immunoassay for extracts of HEK293T cells transfected with MYC-AhR (WT), AhR (1-277;424–848), and HA-STING (WT) by anti-MYC beads. **D** The RMSD values for AhR-STING, AhR-STING-KYN, AhR-STING (dimer), and AhR-STING-KYN (dimer) of the simulation. **E–L** The simulation of the interaction and surface binding model between AhR and STING, including AHR-STING (**E, F**), AhR-STING-KYN (**G, H**), AhR-STING (dimer) (**I, J**), and AhR-STING-KYN (dimer) (**K, L**). AhR is colored with marine, STING with magenta, KYN with yellow. The key residues in AhR are shown as cyan sticks while others as marine stick. The key residues in STING are shown as cyan sticks while others as magenta stick. The red dashes represent hydrogen bond interaction. The blue dashes represent the salt

bridge. **M, N** Coimmunoprecipitation and immunoassay for extracts of HEK293T cells transfected with MYC-STING (140–379), FLAG-AhR (WT) along with HA-UB or its mutants by anti-MYC beads. **O, P** Coimmunoprecipitation and immunoassay for extracts of cell lysate of *AhR* knocked-out cells by anti-STING antibodies. **Q, R** MYC-STING (140–379) or its mutants were individually transfected into HEK293T cells along with HA-AhR by anti-MYC beads. K0 denotes STING with all lysine residues mutated to arginine. **S** Immunoassay for STING expression of K236R cells after indicated treatment in different time points (Mean ± SD). *P* value < 0.05 as statistic difference by two-tailed *t* test. **T** Immunoassay for STING expression of K236R cells after indicated treatment. **U** Coimmunoprecipitation and immunoassay for extracts of K236R cells lysate after treated with Kyn by anti-STING antibodies. **V** Coimmunoprecipitation and immunoassay for extracts of HEK293T cells transfected with MYC-STING (WT), MYC-STING (K236R), and FLAG-TBK1 (WT) by anti-MYC beads after treated with cGAMP. Three biologically independent experiments were performed (**A–C, M–V**). Source data are provided as a Source Data file.

---

of polyubiquitination on STING (Fig. 5O, P and Supplementary Fig. 5F, G).

Then, we conducted another systematic lysine to arginine mutation scanning and found that mutation of the K236 to arginine almost completely abolished ubiquitination induced by activated AhR (Fig. 5Q). Moreover, comparative analysis of the protein sequence of STING between multiple species showed that K236 was a highly conserved lysine site (Supplementary Fig. 5H). To further confirm the involvement of K236, we also generated a STING K0 mutant where all the lysine were converted to arginine. The effect of each site was further investigated by re-introducing lysine to arginine. As expected, the ubiquitination induced by AhR activation was restored after R236 was reversed to lysine (Fig. 5R).

Moreover, we reintroduced WT or K236R-STING into *STING*-KO SYBC1 cells, after verifying that the exogenic mRNA expression of these two forms of STING were comparable at basal levels and under cisplatin treatment (Supplementary Fig. 5I). We further examined the effect of K236R mutation, and found a higher protein level of STING than the unmutated form as well as increased *IFNB1* expression (Supplementary Fig. 5I). To further confirm the role of K236 endogenously, we constructed K236R knock-in mutant SYBC1 cells through method based on Cas9 and oligonucleotide strands[43,44]. Successful constructed mutant was confirmed via sequencing (Supplementary Fig. 5J). Consistent with previous data, endogenous K236R-STING had a longer degradation half-life under cGAMP treatment than wild-type (Supplementary Fig. 5S). At the same time, under the treatment of Kyn, the protein level of K236R-STING was also higher than that of the wild-type and showed decreased ubiquitination levels as well (Fig. 5T, U). Cells expressing endogenous K236R-STING showed higher expression of *IFNB1* than WT cells under cisplatin or cGAMP treatment (Supplementary Fig. 5K). Previous studies indicated that polyubiquitination on STING might also influence its interaction with TBK1[45], but our coimmunoprecipitation assay showed that K236R mutation had no obvious influence on the STING-TBK1 interaction (Fig. 5V). Together, these data suggested that AhR interacted with STING directly and through a switch-on mode to confer a K48-linked polyubiquitin chain at the K236 site.

## AhR worked as an adaptor to facilitate STING ubiquitination through CUL4B-RBX1 E3 complex

It was previously reported that AhR act as an adaptor bridges Cullin 4B E3 ligase complex and substrates[39]. We investigated the function of core components of the E3 ligase complex, CUL4B and RBX1, under our experimental conditions using siRNA. As expected, knocking down *CUL4B* or *RBX1* abolished the effect of AhR or Kyn on STING ubiquitination and extended half-life of STING in degradation (Fig. 6A–D). Furthermore, using an exogenously expressed MYC-tagged STING, HA-tagged AhR, and FLAG-tagged CUL4B, we found that STING

interacted with CUL4B, and the interaction was enhanced by AhR, especially its activated form (Fig. 6E). Moreover, through antibodies towards endogenous molecules, we also confirmed this STING-AhR-CUL4B triangular interaction in the native condition (Fig. 6F). Consistently, the immunofluorescence assays showed that STING and CUL4B were colocalized following cGAMP treatment, and this effect was abolished by AhR deficiency (Fig. 6G).

Furthermore, we constructed human and murine CUL4B (SYBC1)/*Cul4b* (MB49)-KO cell lines and found that tumor cells lacking CUL4B showed enhanced STING stability and IFN-β production under treatment of cisplatin, RT, or cGAMP (Fig. 6H and Supplementary Fig. 6A, B). In addition, compared with wild-type cells, *Cul4b*-KO cells formed significantly smaller tumors in mice (Fig. 6I); when *Sting* was furtherly knocked out in *Cul4b*-KO cells, the double-KO cells formed bigger tumors than wild-type cells (Fig. 6J and Supplementary Fig. 6C). These data suggested that AhR acted as an adaptor in an on-off ligand-dependent switchable mode to bridge the CUL4B-Rbx1 E3 ligase complex and STING.

## SLC7A5 acted as critical Trp transporter to regulate STING stability

Our previous data indicated that Cisplatin treatment led to increased Trp intake. Thus, we screened several transporters including SLC1A5, SLC7A5, SLC16A10, and SLC36A4, which were previously identified to mediate Trp importation[21], and found that knocking down *SLC7A5*, but not other transporters, obviously upregulated STING expression and enhanced IFN-β secretion under cisplatin and cGAMP treatment (Fig. 7A, B and Supplementary Fig. 7A, B). Moreover, we also found that cisplatin or cGAMP treatment upregulated the expression of *SLC7A5* (Fig. 7C), whereas knocking down *SLC7A5* decreased the levels of both intracellular Trp and Kyn (Fig. 7D and Supplementary Fig. 7C). These data suggested that SLC7A5 was critical for enhanced engulfment of Trp after cisplatin treatment.

To further confirm the critical role of SLC7A5, we used CRISPR to construct KO cell lines (Supplementary Fig. 7D), which were then treated with cisplatin or cGAMP. The results showed that *SLC7A5*-KO increased STING protein level, subsequent signaling activation, and IFN-β secretion (Fig. 7E and Supplementary Fig. 7E). In addition, decreased intracellular Kyn and Trp levels were detected in *SLC7A5*-KO cells (Fig. 7F and Supplementary Fig. 7F). We also found that in *SLC7A5*-KO cells, the interaction between STING, AhR, and CUL4B was greatly diminished and ubiquitination levels of STING was reduced (Fig. 7G and Supplementary Fig. 7G).

Moreover, in in vivo assays, we found that knocking-out *Slc7a5* reduced tumor growth (Supplementary Fig. 7H, I); while *Slc7a5/Sting* double-KO cells formed bigger but not statistically significant tumors than those formed by wild-type cells (Fig. 7H and Supplementary Fig. 7J); in line with this, SLC7A5 inhibitor BCH also furtherly reduced

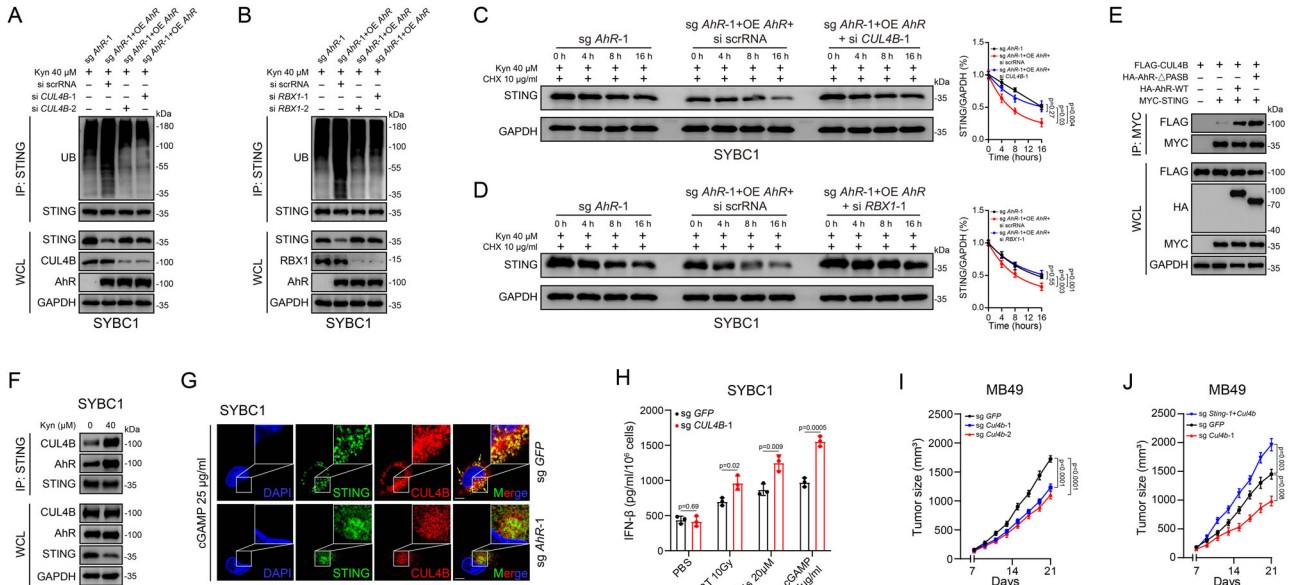

**Fig. 6 | AhR worked as an adaptor to facilitate STING ubiquitination through CUL4B-RBX1 E3 complex. A, B** Coimmunoprecipitation and immunoassay for extracts of cell lysate by anti-STING antibodies in *CUL4B* (**A**) and *RBX1* (**B**) knocked-down cells. **C, D** Immunoassay for STING expression of *AhR* knocked-out cells with overexpressed *AhR* and knocked-down *CUL4B* (**C**) or *RBX1* (**D**) after treated with Kyn in different time points. **E** Coimmunoprecipitation and immunoassay for extracts of HEK293T cells transfected with MYC-STING, HA-AhR (WT), AhR (1–277; 424–848), and FLAG-CUL4B by anti-MYC beads. **F** Coimmunoprecipitation and immunoassay for extracts of cell lysate after treated with Kyn by anti-STING antibodies. **G** Confocal microscopy for STING and CUL4B expression after being treated with cGAMP in *AhR* knocked-out cells. Scale bar 20 μm. **H** ELISA for IFN-β content in the supernatant of *CUL4B* knocked-out cells after indicated treatment. **I** Effect of *Cul4b* knocked-out on MB49 growth (*n* = 6) (Mean ± SEM). **J** Effect of *Sting/Cul4b* knocked-out on MB49 growth (*n* = 6) (Mean ± SEM). *P* value by one-way ANOVA (**C**, **D**, **I**, **J**). *P* value by two-tailed *t* test (**H**). All *p* value < 0.05 as statistic difference. Error bars represent Mean ± SD, unless otherwise indicated. Three biologically independent experiments were performed (**A**–**H**). Source data are provided as a Source Data file.

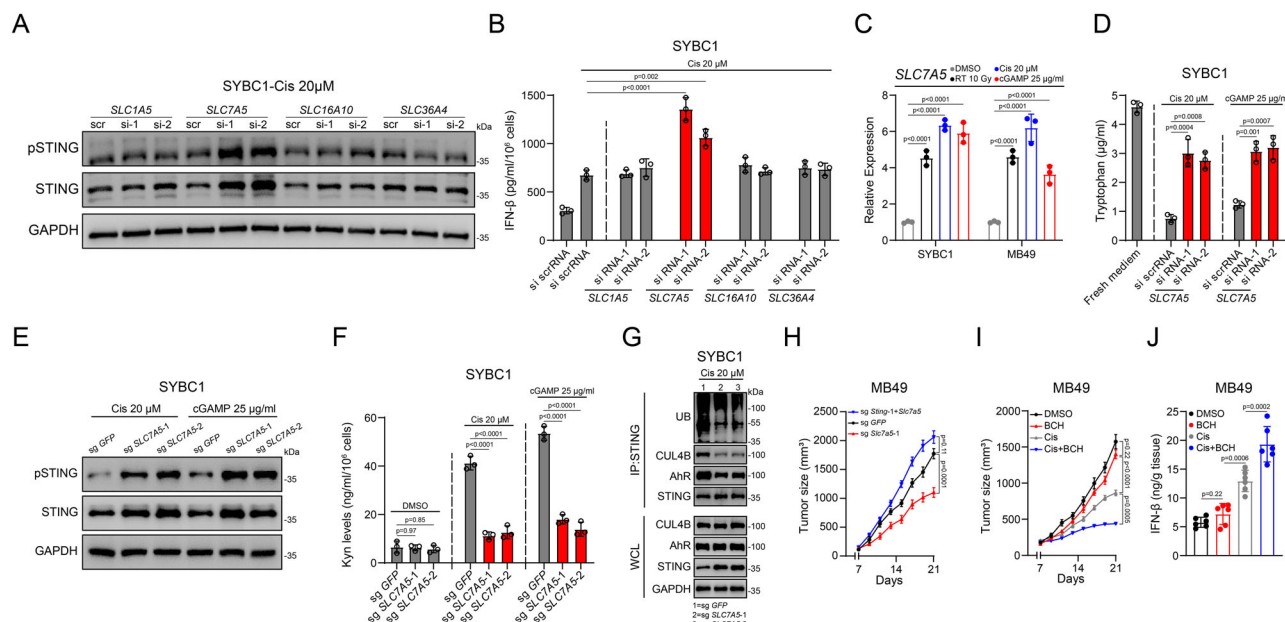

**Fig. 7 | SLC7A5 acted as critical Trp transporter to regulate STING stability. A** Immunoassay for pSTING and STING expression after knocked-down indicated Trp transporters and treated with cisplatin. **B** ELISA for IFN-β content in the supernatant after knocked-down indicated Trp transporters and treated with cisplatin. **C** mRNA expression of *SLC7A5* (SYBC1)/*Slc7a5* (MB49) after indicated treatment. **D** HPLC-MS detection of Trp remaining in the supernatant after knocked down *SLC7A5* and treated with cGAMP or cisplatin. **E** Immunoassay for pSTING and STING expression after knocked-out *SLC7A5* and treated with cGAMP or cisplatin. **F** HPLC-MS detection of Kyn after knocked-out *SLC7A5* and treated with cGAMP or cisplatin. **G** Coimmunoprecipitation and immunoassay for extracts of cell lysate in *SLC7A5* knocked-out cells and treated with cisplatin by anti-STING antibodies. **H** Effect of *Sting/Slc7a5* knocked-out on MB49 growth (*n* = 6) (Mean ± SEM). **I, J** Effect of BCH combined with cisplatin on MB49 growth (*n* = 6) (Mean ± SEM) (**I**) and IFN-β production (**J**) in tumor. *P* value by one-way ANOVA (**B**–**D**, **F**, **H**, **I**, **J**). All *p* value < 0.05 as statistic difference. Error bars represent Mean ± SD, unless otherwise indicated. Three biologically independent experiments were performed (**A**–**G**). Source data are provided as a Source Data file.

tumor growth and increased intratumoral IFN-β under cisplatin or diABZI treatment (Fig. 7I, J and Supplementary Fig. 7K, L). More convincedly, knocking-out *Sting* almost abolished the synergistic effect of BCH with cisplatin (Supplementary Fig. 7M). Furthermore, in the MB49 orthotopic tumor model, we found that either inhibition of AhR, IDO1 or SLC7A5 could achieve synergistic effect with cisplatin (Supplementary Fig. 7N).

## Trp metabolism-AhR-STING pathway affects the efficacy of NAC in BC patients

Next, we investigated whether the clarified mechanism underlying the regulation of IFN-Is by Trp metabolism is clinically relevant, especially in predicting the efficacy of NAC. To this end, we separated patients in Sjodahl et al.'s cohort into responders and non-responders, and found that gene sets including those of Trp metabolism, AhR activation, and Kyn production were upregulated in non-responders (Fig. 8A). In contrast, that cGAS-STING-related gene sets were upregulated in responders (Fig. 8B). Furthermore, STING activation and effector T-cell signatures were highly correlated not only in Sjodahl NAC public datasets[22,23,46] (Fig. 8C); and higher levels

of STING activation predicted better NAC response and longer survival of patients in Sjodahl et al's cohort (Fig. 8D, E). Other than patients in Sjodahl et al's cohort, in Tabar et al's cohort that patients with higher Trp metabolism or stronger Trp transportation signatures also showed worse response to NAC treatment (Fig. 8F, G). The occurrence and development of bladder cancer is closely related to smoking. Tobacco metabolites mainly excrete through urine containing various aromatic hydrocarbon compounds, which can activate AhR[2]. Based on this, we analyzed the continuous smoking and smoking cessation patients in the TCGA database and found that in the smoking cessation patients, AhR showed lower levels of activation, and accompanied by increased IFN-Is, enhanced T-cell infiltration and effectiveness (Fig. 8H–K). Altogether, our data suggested that in patients with BC, Trp-Kyn-AhR pathway suppressed levels of unactivated STING, increasing its threshold for further activation and led to worse NAC response (Fig. 8L).

## Discussion

In this study, we determined that metabolites derived from Trp regulate the protein stability of unactivated STING through the AhR/

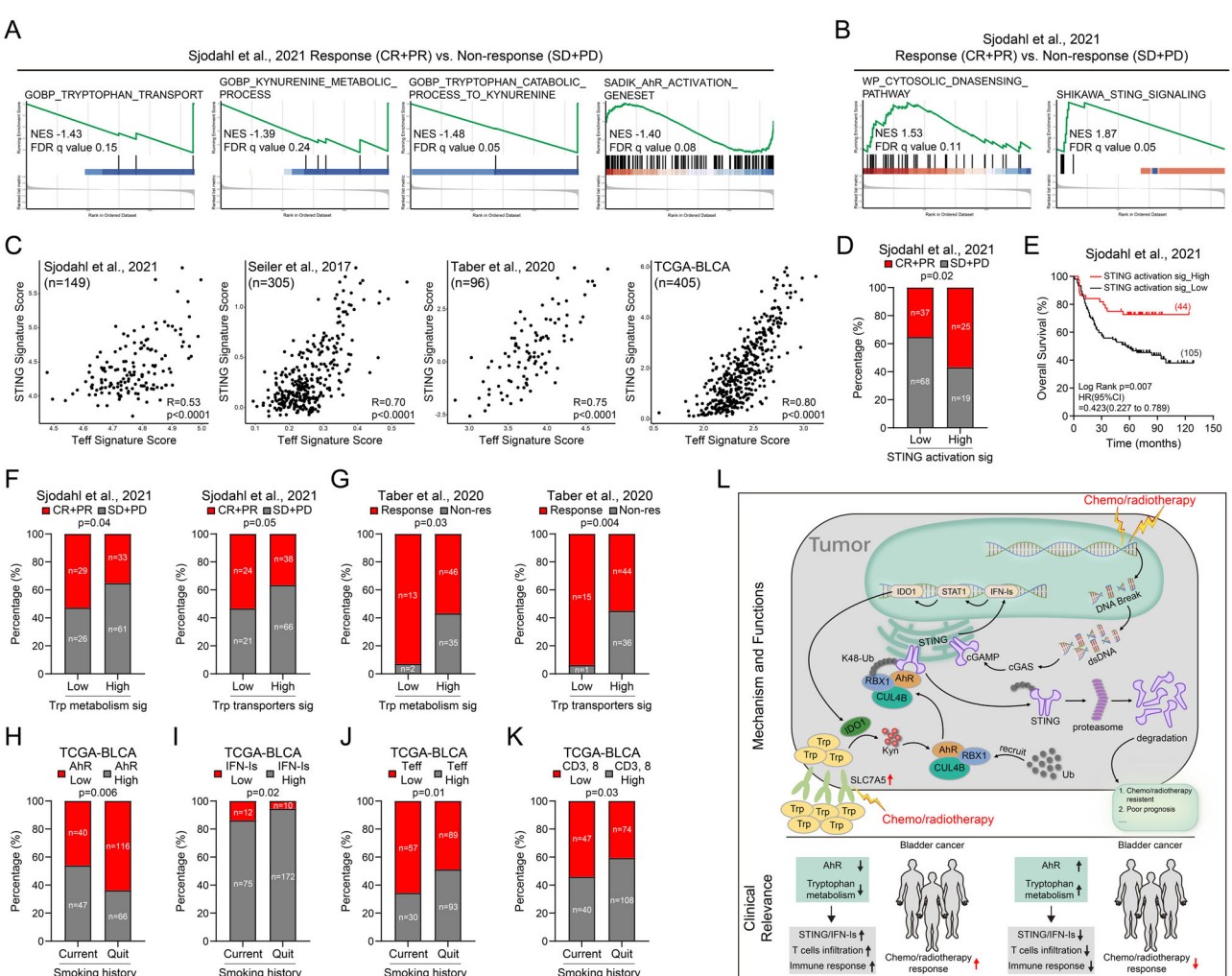

**Fig. 8 | Trp metabolism-AhR-STING pathway affects the efficacy of NAC in BC patients. A, B** GSEA plot shows downregulation and upregulation pathway genes in NAC response samples compared to non-response samples. **C** The correlation between Teff signature expression and STING activation signature expression[22,23,46]. **D** The correlation between NAC response rate and STING activation signature expression[23]. **E** Kaplan–Meier survival curves of patients with low and high STING activation signature expression[23]. **F, G** The correlation between NAC response rate and Trp metabolism signature or Trp transporters signature expression[23,46].

**H–K** The correlation between smoking history and AhR activation (**H**), IFN-Is signaling (**I**), Teff expression (**J**), CD3,8 expression (**K**). **L** The signaling transduction pathways and clinical relevance were illustrated. The correlation coefficient and two-tailed *p* value were calculated using Pearson's correlation analysis (**C**). *P* value by Pearson's chi-square test (**D, F–K**). *P* value by Kaplan–Meier survival analysis (**E**). AUC was used to determine the optimal cutoff. All *p* value < 0.05 as statistic difference. Source data are provided as a Source Data file.

Cul4B/RBX1 E3 complex, thereby affecting the production of IFN-Is and the immune-related effect of chemotherapy. More importantly, we found that this process may be inherent metabolism-dependent negative feedback that exists to restrain overactivated cGAS-STING signaling to avoid autoimmune damage. However, dysregulated tumor cell metabolism or accumulation of exogenous AhR ligands (such as aromatic hydrocarbon compounds in tobacco and hair dyes) in the bladder abnormally enhances this feedback mechanism. This eventually leads to tumor cell escape from IFN-dependent immune surveillance. Identification of this mechanism uncovers unrecognized link mediated by AhR between chemicals and IFN-Is production; reveals a new degradation mode of previously poorly studied unactivated STING and provides potential combination drug targets and biomarkers for existing MIBC chemotherapy.

The concept of targeting amino acid metabolism for tumor therapy was initially based on the vigorous proliferative capacity of tumor cells, which demonstrate a strong demand for amino acids to meet the enhanced proliferation[47]. In fact, most treatments targeting amino acid metabolism simultaneously inhibit the function of immune cells in the tumor microenvironment due to that immune cell share most metabolic pathways with tumor cells[20]. Compared with other amino acids, most free Trp (95%) in the host is metabolized through the Kyn pathway and transduces signaling through various metabolites (such as Kyn, KynA, and cinnabarinic acid [CinA])[30,48]. Previous studies, including ours have demonstrated that inhibition of Trp-related pathways not only diminished tumor metastasis and resistance to apoptosis but also enhanced CD8⁺ T-cell function[31,36,37]. Consequently, in contrast to other amino acids, Trp and therapies targeting its metabolism could largely avoid suppressing antitumor immunity, while targeting tumor cells. However, it is also worth note that symptoms of malnutrition often exist in cancer patients, especially those with advanced tumors. Trp is rich in high-protein foods such as meat, milk, and beans. Restricting these high-protein diets will also have adverse effects on tumor treatment. Therefore, inhibiting key node molecules (eg. IDO1, TDO2, AhR etc.) related to Trp metabolism, would have a better scope of application than directly restricting Trp intake through diet, and patients may be more compliant.

Seven enzymes are currently known to catalyze the metabolism of Trp, and for MIBC chemotherapy, we found that IDO1 showed the most prominent effect. This is mainly because chemotherapy leads to the production of IFN-Is in tumor cells and upregulation of the effect of IFN on IDO1. Consistent with this, other enzymes that metabolize Trp are generally highly tissue- or cell-types specific, which are far less universal than IDO1[30]. The immunosuppressive effect of IDO1 has long been recognized, although the specific underlying mechanism remains controversial[49]. Generally, the immunosuppressive effect of IDO1 has been thought to be mediated by the regulation of T-cell function; however, this study revealed an unrecognized link between IDO1 and the cGAS-STING pathway. Moreover, activation of STING also leads to secretion of proinflammatory cytokines such as interleukin-6 (IL6) and tumor necrosis factor (TNF)-α[15,16], suggesting that IDO1 could also exert an anti-inflammatory effect by inhibiting both cytokines. In tumors, the expression of amino acid transporters is often enhanced because of aberrant expression or activation of oncogenes, resulting in increased Trp uptake[30,50]. Moreover, epigenetic modifications and changes in chromatin structure make tumor cells more likely to upregulate IDO1 than normal cells in response to IFN signals[49,51]. Therefore, tumor cells might abnormally amplify the negative feedback regulatory mechanism of IDO1-AhR-STING, which inhibits immune overactivation under physiological conditions, resulting in tumor immune escape.

Chemotherapy can activate antitumor immunity, which plays an important role in the long-term maintenance of its efficacy[52]. The increasing clinical use of immunotherapies targeting immune checkpoints has contributed to cotreatments with chemotherapy and immunotherapy gradually emerging as first-line treatments for multiple tumors[4,6]. In the present work, we further explored the key pathways that regulate IFN-Is production in the presence of chemotherapy and determined the important role of the cGAS-STING pathway in MIBC. Because they can recognize free cytosolic dsDNA, a variety of chemotherapeutic drugs (such as poly-ADP ribose polymerase [PARP] inhibitors, ataxia telangiectasia mutated [ATM] inhibitors, and DNA crosslinking agents) that induce fragmentation of genomic DNA activate cGAS-STING signaling[7,17]. The efficacy of these chemotherapeutics has also been confirmed to be highly related to this activation. However, the effect of cGAS-STING goes beyond these actions and is occasionally paradoxical. For tumor cells in particular, activation of cGAS-STING pathway may also lead to different outcomes. For example, in triple-negative breast cancer, lung adenocarcinoma, and non-small cell lung cancer, activation of cGAS-STING promotes distant or brain metastases and leads to poor patient outcomes[53,54]. The main reason for this contradiction is that in addition to the production of IFN-Is, activation of cGAS-STING also induces nuclear factor (NF)-κB signaling, autophagy, or production of cytokines such as IL6 and TNF-α. These factors can promote tumor stemness or metastatic ability, leading to tumor progression. Our experimental and clinical results suggest that cGAS-STING signaling-derived IFN-Is promote chemotherapeutic efficacy, but strategies to avoid the tumor-promoting disadvantages this pathway remain a challenge.

Post-transcriptional modification of STING plays an important role in regulating its function[15]. Among numerous possible post-transcriptional modifications, ubiquitination of STING is involved in the regulation of its binding to and activation of downstream proteins TBK1 and IRF3, and its oligomerization and protein stability[45,55,56]. Specifically, this complex regulation is mainly achieved through the modification of different lysine sites with distinct ubiquitin chains. Furthermore, previous studies have also shown that ubiquitin modification of STING varies greatly between different types of cells and stimuli. In this study, we found that in the presence of chemotherapy, the Trp-dependent AhR/CUL4B-RBX1 E3 ubiquitin ligase complex mediates ubiquitination modifications on the K236 site of STING with a K48-linked ubiquitin chain and reduces STING protein stability. Compared with other lysine sites, the K236 site shows high conservation. Consistent with our results, USP44-mediated removal of the K48 ubiquitin modification at K236 on STING has been reported to promote STING protein stability following viral infection and enhance antiviral IFN responses[57]. This further suggests the universality and importance of this modification in regulating STING in innate or antitumor immune responses. It is also worth mentioning that, from the perspective of the cell, when STING is activated, there must be two states of STING in the cell, activated and unactivated. A recent study showed that the SEL1L–HRD1 protein complex regulated the stability of unactivated STING[58]. Our study further revealed an intrinsic inhibitory mechanism by which activated STING continuously suppressed the expression level of unactivated STING through downstream signaling. It is further suggested that the dependence of lysosome-dependent protein degradation and proteasome-dependent protein degradation may be different in regulating STING levels of different functional states.

AhR is known to regulate gene expression through its transcriptional function[38]. Our present study revealed that the regulation of STING by AhR does not depend on its canonical transcriptional function, but is mediated through its atypical function as a selective adaptor for E3 ubiquitin ligase. Interestingly, the function as an E3-selective adaptor is also ligand-dependent. Previous studies have shown that the transcriptional and E3 adaptor functions of AhR may be mutually antagonistic under the action of ligands, although the mechanism is unclear. The nuclear translocation of AhR under the action of ligands mainly depends on ligand-mediated allostery and

exposure of the nuclear entry sequence[59]. Our experimental data indicated AhR bound STING mainly thorough C-terminus and ligand binding facilitates this protein interaction. Detailly, our macromolecular calculations have shown that the hidden C-terminus (AD) is extended to grab STING after AhR binds to the ligand. Previous studies demonstrated that a nuclear import sequence that mediates the importation of AhR also located in the C-terminus. Therefore, we speculate that some proteins may bind to the C-terminus of AhR to block its nuclear entry sequence, and thereby prevent the nuclear translocation of AhR. This process might lead to the phenomenon of mutual antagonism of the dual roles of AhR as a transcription factor or E3 ubiquitin-selective adaptor. Collectively, these data suggests that AhR is the key molecule that converts the Trp-Kyn metabolism or exogenous chemical compounds signaling into a protein-protein interaction, which suppresses STING expression in a ubiquitin-proteasome-dependent manner and exerted an immunosuppressive effect.

## Methods

### Patients and tissue samples
All samples were obtained from the Sun Yat-sen University Cancer Center, Sun Yat-sen University, Guangzhou. All patients were pathologically diagnosed with bladder cancer and collected with informed consent, and all experimental procedures were approved by the Internal Review and Ethics Boards of Sun Yat-sen University Cancer Center. RDD number: B2023951478.

### Animal experiments
For the in vivo part of the study, 6–8-week-old female C57BL/6 mice were purchased from the Guangdong Medical Laboratory Animal Center (Foshan, China). These animals were maintained under defined conditions at the Animal Experiment Center of Sun Yat-Sen University, and all animal experiments were approved by the Animal Care and Use Committee of Sun Yat-Sen University. The maximum diameter of tumor mass allowed by ethics committee was 2 cm and was not exceeded in our experiments. For subcutaneously bladder cancer models, C57BL/6 mice were inoculated subcutaneously with $1 \times 10^6$ MB49 cells into the right flank. After tumor growth up to indicated sizes, mice were randomized into different groups on the basis of similar tumor size and body weight for further experiments. For orthotopic tumor model, $1 \times 10^6$ MB49 cells were inoculated into bladder and retained for 1 h, mice were randomized into different groups for further experiments after 7 days. Transferred mice bearing MB49 tumor were treated with cisplatin (5 mg/kg per mouse), CH223191 (100 μg per mouse), StemRegenin 1 (100 μg per mouse), BCH (180 mg/kg per mouse), 1-MT (400 mg/kg per mouse), or diABZI (5 mg/kg per mouse) once every 3 days for 3 times. For Trp-free feed model, Custom feed were purchase from Dyets. Trp deprivation began on day 7–10 after inoculation, relevant treatment was initiated two days after Trp deprivation.

### Cell line culture
SYBC1 were in-house established[26], UMUC-3, MB49, MCF-7, HEK293T and Hela cell lines were obtained from American Type Culture Collection. SYBC1, UMUC-3, MB49, MCF-7, and Hela cell lines were cultured in 1640 with 10% FBS. HEK293T were cultured in DMEM with 10% FBS.

### Single-cell suspension preparation
Bladder cancer samples were processed immediately after being obtained from cystectomy. Every sample was washed with phosphate-buffered saline (PBS) and cut into small pieces (<1 mm³) and transferred into 5 ml DMEM medium containing collagenase IV (1 μg/ml), and subsequently incubated for 60 min on a 37 °C shaker. Subsequently, 4 ml PBS was added to dilute the suspension, and then a 70-

μm cell mesh was used to filter the suspension. After centrifugation at 800 rpm for 5 min, we collected the cell pellet and resuspended it with cell preservation liquid.

### Pathway analysis and signature
Gene set enrichment analysis (GSEA) were conducted with the R package clusterProfiler (version 3.12.0) based on gene expression matrice. Pathway lists were provide by public databases MSigDB (http://www.gsea-msigdb.org/gsea/msigdb). Signature genes and pathways were listed in Supplementary Table 6.

### Analyses based on public datasets
TCGA datasets obtained from UCSC Xena (https://xena.ucsc.edu/). To evaluate the effect of therapeutic response, BLCA cohort with neoadjuvant chemotherapy were analyse[22,23,46].

### De novo modeling
The protein structure of AhR is predicted by the I-TASSER server[60], which is an online resource for automated protein structure prediction and structure-based function annotation (https://zhanggroup.org/I-TASSER/). It is a hierarchical template-based method. I-TASSER Results for AhR refer to Supplementary Data 1.

### Molecular docking
The structure of the STING was downloaded from AlphaFold[61] and the ID is AF-Q86WV6-F1-model_v2. Protein-protein docking in ClusPro 2.0[62] was used for molecular docking simulation for AhR with STING. For protein docking, STING is set as ligand and AhR as receptor. The ligand was rotated with 70,000 rotations. For each rotation, the ligand was translated in $x$, $y$, and $z$ axis relative to the receptor on a grid. One translation with the best score was chosen from each rotation. Of the 70,000 rotations, 1000 rotation/translation combinations that have the lowest score was chosen. Then, a greedy clustering of these 1000 ligand positions with a 9 Å C-alpha RMSD radius was performed to find the ligand positions with the most "neighbors" in 9 Å, i.e., cluster centers. The top ten cluster centers with most cluster members were then retrieved and inspected visually one by one. The pose with the most intermolecular contacts were finally identified as the best probable binding mode between AhR and STING. Dock module in MOE (Molecular Operating Environment v2018.01)[63,64] was used for molecular docking of small molecule KYN and AhR-STING complex. The 2D structures of the KYN was drawn in ChemDraw 18.2 and converted to 3D structures in MOE v2018.01[64] through energy minimization, as ligands. Prior to docking, the force field of AMBER10: EHT and the implicit solvation model of Reaction Field (R-field) were selected. The PAS-B domain of AhR was selected as ligand binding pocket. The "induced fit" protocol was selected, in which the side chains of the binding site in receptor were allowed to move according to ligand conformations, and a constraint was applied on their positions. The weight used for tethering side chain atoms to their original positions was 10. Firstly, all docked poses were ranked by London dG scoring function, then force field refinement was applied on the top 30 poses followed by a rescoring of GBVI/WSA dG scoring function. The conformation with the lowest binding free energy was finally identified as the best probable binding mode between AhR-STING complex and small molecule KYN.

### All-atoms molecular dynamics (MD) simulations
Since our purpose of simulations were to describe the effect of the KYN molecule on binding between AhR and STING which was just a problem at the atomic level, obviously high accuracy of quantum mechanics (QM) at the electron level used to investigation the small molecule system and low accuracy of coarse-grained MD at the mesoscale level

used to investigation the disperse system were not suitable for our systems. Therefore, all-atoms MD simulations with medium accuracy were performed with GROMACS (version 2020.6)[65] for the complex of AhR-STING, AhR-STING-KYN, AhR-STING (dimer) and AhR-STING-KYN (dimer) (Supplementary Table 7). Before the MD simulations, the protonation states of the charged residues (His, Asp, Glu, Lys and Arg residues) were firstly determined by the H++ program and careful examination of their individual local hydrogen-bonding networks[66]. For His residues, H832 residue in AhR was determined as doubly protonated on ε site and δ site, H326/H394/H511/H616/H618/H679/H780/H826 residues in AhR and H72 residue in STING were determined as singly protonated on ε site, and H39/H155/H163/H175/H247/H291/H337/H527/H555/H625/H626/H644/H711/H750/H782/H831 residues in AhR and H3/H7/H16/H42/H50/H74/H157/H185/H232/H332 residues in STING were determined as singly protonated on δ site. For other charged residues, such as Asp, Glu, Lys and Arg residues, the default protonation states (charged states) were determined. Meanwhile, AMBER14SB[67] Force Field and General AMBER Force Field (GAFF)[68] parameters were used for the protein and KYN molecule, respectively. The partial atomic charges of KYN was calculated by the restrained electrostatic potential (RESP)[69] charge following the optimization of KYN at B3LYP/6-31 G(d) level by Gaussian 16[70] package. The complexes were neutralized by adding neutralizing ions (0.15 mol/L NaCl) and solvated in a box with TIP3P[71] explicit water molecules. The solvent layers between the box edges and solute surface were set to 1.2 nm. The particle mesh Ewald (PME)[72,73] method was employed to treat the long-range electrostatic interactions and the calculated cutoff value of van der Waals interaction was 1.2 nm. During the MD simulations, minimization, equilibration runs and production runs were proceeded in turn. For the minimization, the systems were relaxed by 1000 steps using the steepest descent algorithm followed by other 1000 steps using the conjugate gradient method. For the equilibration runs, the temperature and the pressure were controlled by using the Berendsen coupling algorithm with a time constant of 0.1 ps and 1.0 ps, respectively[74]. And the proteins and KYN molecules were constrained for the sake of the relaxation of water molecules in 100 ps. For the production runs, an integration time step of 2 fs was employed to integrate the equations of motion, and the Parrinello-Rahman coupling algorithm was used to keep the pressure constant[75]. Meanwhile, the simulated temperature was set to 298.15 K and 500 ns molecular dynamics simulations was performed in the NPT ensemble. The structures in the production runs were abstracted with an interval of 100 ps (in total of 5000 structures for every system) to calculate the root-mean-square-deviation (RMSD) tendency of the heavy atom of proteins, which were used to split the production runs into equilibration phase and production phase, and the time stage in which the RMSD values have stabilized was considered to be the production phase (last 200 ns was considered as the production phase of the monomer and dimer of AhR-STING complex and last 400 ns was considered as the production phase of the monomer and dimer of AhR-STING complex). Since all the systems have reached a stable state in a very short time (at most 300 ns) based on the RMSD tendency, the enhanced sampling method was not need to consider in our simulations. The structures in the production phase were abstracted also with an interval of 100 ps (in total of 2000 structures for each of monomer and dimer of AhR-STING complex and in total of 4000 structures for each monomer and dimer of AhR-STING-KYN complex) to calculate the RMSF tendency of the protein residues, analysis the interactions between AhR and STING, and obtain the binding free energy between AhR and STING. The binding free energy between AhR and STING was calculated with gmx_MMPBSA (version 1.4.3)[76] based on MMPBSA.py[77] from AmberTools 20 suit. The molecular visualizations of the dynamic movie were carried out using the VMD v1.9.4 (Visual Molecular Dynamics) software[78], and 3D interaction diagrams of complex were created with MOE v2018.01 program[64].

## CRISPR genome editing

To generate indicated knocked-out cells, optimal Single-guide RNA (sgRNA) target sequences were designed using Benchling. sgRNA target sequences were cloned into the PX458 plasmid. SYBC1 and MB49 were transfected and sorted, and the knock-out efficiency was validated by western blot. The sequences of the primers are listed in Supplementary Table 8.

Genomic mutations were introduced into cells using the CRISPR–Cas9 system, as described previously[43]. sgRNA were designed to target the genomic area adjacent to mutation sites in STING (K236R) using the CRISPR design tool (https://www.benchling.com/). The annealed guide RNA oligonucleotides were inserted into a PX458 vector (Addgene) digested with the BbsI restriction enzyme[44]. Cells were seeded at 60% confluence, followed by co-transfection of sgRNAs (0.5 μg) and single-stranded donor oligonucleotide (ssODN) (10 pmol) as a template to introduce mutations. Twenty-four hours after transfection, GFP-positive cells were sorted via flow cytometry, diluted for single cells, and seeded into 96-well plates. After cell amplification, genomic DNA was extracted, followed by sequencing of the PCR products spanning the mutation sites. The lower-case letters in the ssODN sequences indicate the mutated nucleotides that will replace the endogenous nucleotides in the genomic DNA of parental cells using the CRISPR–Cas9 system. sgRNA targeting sequence and PCR products amplified from the following primers are listed in Supplementary Table 8.

## Short interfering RNAs transfer

The short interfering RNAs (siRNAs) targeting indicated genes were designed and synthesized by GenePharma (Shanghai, China). RNAiMAX (Invitrogen) was used to transfect according to the manufacturer's instructions, and knock-down efficiency was validated by qPCR or western blot. The sequences of the primers are listed in Supplementary Table 8.

## Flow cytometry

BC samples or transferred mice bearing MB49 tumor were harvest and digested into single-cell suspension for further analyse. For cell surface marker analysis, all samples were stained with Live/Dead dye for 15 min at room temperature before they were stained with different antibodies. Cells were resuspended in PBS containing 1% FBS and stained with indicated fluorescent-conjugated antibodies for 30 min at 4 °C. Intracellular Fixation & Permeabilization Buffer Set (Invitrogen) was used according to the manufacturer's instructions. All analyses were conducted by cytoFLEX LX. The antibodies used are listed in Supplementary Table 9.

## Immunofluorescence

For immunofluorescence, cells were fixed, permeabilized, blocked. By using a PANO 7-plex kit according to the manufacturer's instructions, samples were incubated with dilutional primary antibodies. DAPI was then used to counter the nuclei and images obtained by laser scanning confocal microscopy (LSM880, Zeiss). The antibodies used are listed in Supplementary Table 9.

## Radiation model

C57BL/6 mice were inoculated subcutaneously with $5 \times 10^5$ MB49 cells into the right flank. When tumor volume (determined by length × width$^2$/2) reached $100 \pm 20$ mm3, tumors were irradiated with 21 Gy (MB49) or 10 Gy (SYBC1, UMUC-3) from a 225-kVp cabinet X-ray irradiator filtered with 0.5-mm Cu (IC-250, Kimtron).

## ELISA

SYBC1, UMUC-3 cells were cultured in 1640 with 10% FBS with indicated confluency. The supernatant or cells were harvested and used for subsequent ELISA assay. For tumor tissue from mice, samples were

weighed and extracted using Tissue Protein Extraction Reagent (ThermoFisher Scientific). All experiments were performed according to the manufacturer's instructions.

## High-performance liquid chromatography with q-exactive mass spectrometry

To measure the level of Kyn and Trp, cells or tissues were washed twice in PBS and lysed in extraction solvent (80% methanol/water) for 30 min at −80 °C. After centrifugation at 12,000 × $g$, 10 min at 4 °C, supernatant extracts were analyzed by HPLC-MS. Supernatant were extraction, filtration, drying and re-solution for further HPLC-MS.

## Western blot and immunoprecipitation

For immunoprecipitation, whole-cell extracts were acquired after transfection or stimulation with appropriate ligands, followed by incubation with anti-myc beads or the appropriate antibodies plus Protein A/G (Pierce). Beads were then washed five times with low-salt lysis buffer, and immunoprecipitates were eluted with 2× SDS Loading Buffer (Cell Signaling Technology) and resolved by SDS–polyacrylamide gels, and then transferred to NC membranes. Primary antibodies against indicate genes and anti-myc beads were used according to manufacturer's instructions. Peroxidase conjugated secondary antibody (CST) was used, and the antigen-antibody reaction was visualized by enhanced chemiluminescence assay (ECL, Thermo). The antibodies used are listed in Supplementary Table 9.

## Two-step immunoprecipitation and ubiquitination assays

Two-step immunoprecipitation and ubiquitination assays were performed according to the previous report[79]. For the first-step immunoprecipitation assay, whole-cell extracts were prepared by using low-salt buffer supplemented with a 5 mg/ml protease inhibitor cocktail (Roche). Lysates were incubated with the anti-FLAG beads overnight. The immunoprecipitates were washed 3–5 times with low-salt buffer. For the second-step immunoprecipitation assay, the immunoprecipitates were denatured by boiling for 5 min in the Lysis buffer containing 1% SDS. The elutes were diluted 1:10 with low-salt buffer. The diluted elutes were re-immunoprecipitated with anti-MYC beads overnight. After 3–5 times washing, the immunoprecipitates were resolved by SDS-PAGE. The antibodies used are listed in Supplementary Table 9.

## RT-qPCR

RNA was extracted from indicated cells, and reverse transcription of the first-strand cDNA was performed using a reverse-transcription kit (Promega). The RT-qPCR assay was conducted on the Bio-Rad SPX (96 or 384) system with a 2× SYBR Green mix (Life, Carlsbad, CA, USA). The data were normalized to the expression of GAPDH. The sequences of the primers are listed in Supplementary Table 8.

## Luciferase reporter assay

Indicated cells were seeded in 12-well plates for 12 h and co-transfected with pGL3-IFNB1 reporter plasmid and a phRL-TK-Renella luciferase control vector (Addgene). After 24 h, indicated reagents were added, the cells were lysed in 1× passive lysis buffer (Promega) for 24 h, and the luciferase activities were determined using the Dual-Luciferase® Assay kit (Promega). Firefly luciferase activity was normalized to Renilla luciferase activity for each sample.

## Statistical analysis and reproducibility

Data are presented as the mean ± SEM or mean ± SD of at least three independent experiments. Statistical analyses were performed using either GraphPad Prism9 (GraphPad) or SPSS Statistics version 24 (IBM). Two-tailed Student's $t$ test, two-tailed Wilcoxon test, and one-way ANOVA with Tukey's test for multiple comparisons were used to calculate $P$ values. Time-to-event data were described using Kaplan–Meier curves, and differences in survival were determined using the log-rank test. The two-tailed Pearson's chi-square test or Pearson's correlation analysis was used to compare clinical characteristics. A $P$ value of <0.05 was considered statistically significant. Three biologically independent experiments were performed unless otherwise stated.

## Reporting summary

Further information on research design is available in the Nature Portfolio Reporting Summary linked to this article.

## Data availability

TCGA datasets for BLCA, were all obtained from UCSC Xena [https://xenabrowser.net/datapages/]. BLCA cohort with neoadjuvant chemotherapy datasets were obtained from Gene Expression Omnibus (accession numbers: GSE169455 and GSE87304) or as supplementary material to the article (https://www.nature.com/articles/s41467-020-18640-0)[46]. I-TASSER results for AhR refer to Supplementary Data 1. Initial and final configurations for molecular dynamics trajectories refer to Supplementary Data 1. The remaining data supporting the findings of this study are available within the Article, Supplementary Information, or Source Data file. Source data are provided in this paper.

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

## Acknowledgements

This work was supported by grants from the National Natural Science Foundation of China (82073103, 82203320, 82002666, and 82103449); 308 Project of Sun Yat-sen University Cancer Center; Sun Yat-sen University Clinical Research 5010 Program (2019011); Post-doctoral Innovation Talent Support Program (BX20200396); China Postdoctoral Science Foundation funded project (2021M693644).

## Author contributions

Z.K.M., X.Y.L., J.C., and Z.W.L. conceived the ideas and designed the experiments. Z.K.M., Z.Y.L., Z.F.L., Y.Z.W., and J.W.Y. performed the experiments. C.W. performed bioinformatics data analysis. Z.K.M., C.W., and Y.Z.M. performed clinical data analysis. Z.K.M., Z.Y.L., Y.Z.M., J.C., and X.Y.L. analyzed the data. Z.K.M., X.Y.L., and Z.W.L. wrote the paper. J.C., Z.W.L., and X.Y.L. jointly supervised this work. All authors reviewed and discussed the final version of the paper.

## Competing interests

The authors declare no competing interests.
