## [Peer Review File · Nature Communications]

AhR Diminishes the Efficacy of Chemotherapy via Suppressing STING Dependent Type-I Interferon in Bladder CancerEditorial Note: Parts of this Peer Review File have been redacted as indicated to maintain patient confidentiality.

REVIEWER COMMENTS

Reviewer #1 (Remarks to the Author): with expertise in molecular dynamics

In this study, the authors used diverse experimental techniques to systematically investigate the regulation of IFN- β production under chemo drugs treatment in bladder cancer (BC) cells through the role of amino acids and then the role of AhR. In conclusion, through the systematic experiment, they described how the Trp-IDO1-Kyn metabolic pathway influences IFN- β production. Eventually, they identified that Trp metabolism-AhR-STING pathway could affect the efficacy of the treatment. While the research is certainly important to understand the mechanism, the study was well conducted. Some comments are given below to enhance the manuscript further.

Page 14 of the manuscript (PDF version), "In addition, Trp deprivation or knocking-out IDO1 also decreased the ubiquitination of STING, while this could be reversed by adding Kyn (Figures 4E, S4D-E)." Readers could misunderstand this sentence. I recommend saying your result more explicitly.

Page 17 of the manuscript, regarding molecular docking and MD simulations, the question is whether STING exists as a homodimer in the transmembrane domain of ER. However, the authors used the monomer of STING for molecular docking and MD simulation. The author could elaborate on how the monomer interacting with AhR in the study could elucidate the interaction between AhR and homodimer in nature. The question is whether the AhR binds the monomer differs from how the AhR binds the dimer.

Page 18 of the manuscript says, "Then, based on Molecular Mechanics Poisson-Boltzmann Surface Area (MMPBSA), we found that the bind free energy upon AhR with STING for AhR-STING and AhR-STING-Kyn was shifted from -127.36 kcal/mol to 171.43 kcal/mol." It was a huge change in the binding energy between two systems "from -127.36 kcal/mol to 171.43

kcal/mol.” Please confirm this observation and calculations. Right now, in AhR-STING-Kyn system, its binding energy is positive. So, the binding is not stable.

Also, ideally, more simulations should be performed to draw a more robust conclusion. I understand the difficulty of doing extra calculations; however, the authors must address this issue properly and justify their analysis.

Pages 18 and 19, regarding ubiquitinations by K48 and K236. One part indicates that ubiquitinations reappeared when only K48 was reintroduced in the KO mutation. Another part says, “the ubiquitination induced by AhR activation was restored after only R236 was reversed to lysine.” So, I could read that K48 is no longer necessary if AhR induces ubiquitination. What would be the insights from the mutation experiment for these two lysine residues?

Minor issues:

On Page 17, the manuscript says, “We found that the transcription activation domain-mediated the interaction with STING, rather than the PAS region that was previously reported to interact with other proteins (Figure 5B).” Please cite appropriate references

Figure S2 – Typo in the description

Figure S5 (F) says, “Alignment of cGAS amino acid sequences.” This must be a typo. It should say, “Alignment of STING amino acid sequences.”

Reviewer #3 (Remarks to the Author): with expertise in STING

In this study, Ma et al. found that AhR acted as an adaptor that linked STING to Cul4B/RBX1 E3 ligase for proteasome degradation during cisplatin treatment in bladder cancer. Trp-IDO1-Kyn metabolic pathway regulated AhR-mediated STING ubiquitination and degradation. Pharmacologic inhibition of this pathway also increased STING stability and antitumor efficacy of cisplatin treatment in bladder cancer. Overall, this is an interesting study. Some concerns listed below need to be addressed properly.

1. Previous studies showed that STING activation-induced degradation was mainly controlled by lysosome, and proteasome played a minimal role during this process (PMID: 29241549). However, the current study showed that AhR-mediated proteasome degradation played a major role in controlling STING stability. To reconcile this discrepancy, the authors must demonstrate which pathway play a dominant role in regulating STING degradation during cisplatin treatment in the bladder cancer cells. This can be done by treating the cells with Cisplatin/cGAMP followed by MG132 or CQ treatment.
2. In the Figure 1, does Trp affect IFN induction by other stimuli, such as DNA, RNA or LPS?
3. It has been known that STING undergoes lysosome-mediated degradation after activation. However, in the Figure 2D and 2H, STING protein level was not decreased, rather it was increased by cisplatin or RT or cGAMP treatment. These data seemed to be in conflict with previous studies and should be verified. If so, can author comment on why STING was degraded after cGAMP treatment in those conditions?
4. In the Figure 3, was the antitumor effect of IDO1 inhibitor treatment dependent on STING? The 1-MT treatment should also be tested on STING KO cells. Similar for the Figure 4M and 4P, STING KO tumor model should also be tested with AhR inhibitors.
5. In Figure 5, endogenous STING-AhR interaction should also be examined by immunoprecipitation. Whether their interaction was affected by Trp or kyn? K236 looked like the major ubiquitination site by AhR to induce STING degradation. However, all the experiments were performed with STING and AhR overexpression system. Whether K236 regulated STING degradation in the native condition should be examined. K236R CRISPR knock in cell line should be generated and tested for STING degradation and stability.
6. In the Figure 6, The E3 ligase Cullin 4B should have many other substrates that may also affect antitumor immune response. Whether the antitumor effect observed in Cullin 4B KO cells was dependent on STING? The authors should compare tumor growth between Cullin 4B KO and Cullin 4B/STING double knock out cells. Similar in the Figure 7, the control experiments were not well designed. The tumor growth should be monitored in parallel between SLC7A5 KO cells and SLC7A5/STING double KO cells. STING KO tumors should also be tested for the synergetic antitumor effect of Cis+BCH.

Reviewer #4 (Remarks to the Author): with expertise in IDO-Kyn-AHR, cancer

The study “AhR Diminishes the Efficacy of Chemotherapy via Suppressing STING Dependent Type-1 Interferon in Bladder Cancer” by Ma, Z. et al. identifies a novel mechanism by which AhR decreases STING levels through physical interaction with STING. The authors demonstrate that both genetic and pharmacological inhibition of AhR caused an increase in STING expression and correlative IFN-I levels. Furthermore, the study implicates this finding as an oncogenic mechanism through which bladder cancer (BC) cells can repress IFN-I levels which would otherwise stimulate antitumor immunity. This study is significant not only due to the characterization of a novel contributor to immune evasion, but due to the discovery of a physical connection between the kynurenine and STING pathways. However, additional work and clarifications are necessary to more thoroughly establish the major findings of the paper.

Major comments:

1. The authors indicate that, in BC, tumor cells are responsible for the majority of IFN-I s in the tumor microenvironment. However, they only analyzed 4 out of 13 IFN-I s. Given that myeloid-derived cells, particularly dendritic cells, are classically considered as primary contributors to IFN-I s, authors should either clarify that tumor cells are responsible for the majority of those specific IFN-I subtypes or expand their study to include a majority of IFN-I s.
2. Furthermore, regarding the BC patients the cells were isolated from, there is no mention of what therapeutic interventions, if any, the patients were undergoing as that may contribute to the status of IFN-I concentrations.
3. In Figure 1, the authors demonstrate that Trp-deficiency along with cis-plat treatment increased CD8+ T cell infiltration. However, the experiment was relatively short at 3 weeks. While the field has moved away from the thought that Kyn-AhR tolerization is driven through Trp deprivation, it is unclear if Trp deprivation would not become a concern in Trp-starved patients. Therefore, authors should address this in their Discussion.
4. Can Kyn decrease STING at lower concentrations? 400uM Kyn is well beyond the amount necessary to promote AHR translocation, which following the proposed model, should be the driver decreasing STING. STING decrease should be visible between 10-40uM of Kyn. 400uM Kyn would be cytotoxic to most cells. Therefore, authors must repeat experiments using lower amounts of Kyn and verify that the same effects are still observed for all findings. Effects of Kyn in cell growth, proliferation, Ahr translocation and target gene

activation also must be verified.

5. Given that IDO1 is implicated as responsible for the activation of AhR along with the antitumoral phenotype being CD8+ T cell driven, it is surprising that the authors do not measure or mention IFN-g. While an IFN-II, IFN-g is the best characterized inducer of IDO1 and is an inflammatory cytokine secreted by CD8+ T cells. How does cis-plat induction of IDO1 compare to IFN-g? Do IFN-g levels change similarly with IFN-Is?

6. Do high-levels of Kyn, IDO1, and AHR correlate with cis-plat resistant BC tumors? Furthermore, do STING levels follow the expected trend?

Other comments:

1. The ending of the abstract should be re-written for clarity.
2. "Adoptive immunity" presumably should be adaptive immunity.
3. Authors need to clarify that IDO1 is not the sole rate-limiting enzyme.

Reviewer #5 (Remarks to the Author): with expertise in immuno-metabolism, AhR
The report is an interesting and novel exploration of the role for IFN-driven feedback that may promote immune suppression following treatment with platinum-based chemotherapy. The main strength of the report is the novel link identified between non canonical AhR function and reduction of STING signaling that may reduce IFN production and immune-mediated reduction of tumor burden in bladder cancer. Overall the experiments are well designed and thorough, describing a function interaction that results in ubiquitin-mediated reduction of STING and attenuation of STING signaling. However there are some significant points that somewhat diminish enthusiasm for what is otherwise an interesting report.

Major points-

- 1- The manuscript relies entirely on in vivo results with one bladder cancer cell line, MB49. The authors should show the applicability across multiple models of bladder cancer using spontaneous models, if possible.
- 2- In figure 1k mice were placed on Trp-free chow for 21 days. It is surprising that mice could tolerate the loss of Trp for this extended amount of time as extended loss of Trp can cause

cellular/tissue pathology, behavioral/cognitive issues, and eventual mortality. It would be appropriate to address this caveat in the description of the experiment.

3- Despite the fact that the main hypothesis proposes that AhR limits immune-responses post therapy, there is a complete lack of immune characterization of the tumors +/- Trp, +/- AhR inhibition, or +/- IDO inhibition. This should be done to demonstrate that alterations in the immune infiltrate composition or activation state accompany alterations to the aforementioned pathways to further strengthen the argument.

4- IFN may impact the tumors directly altering immunogenicity, MHC expression, etc.. Alternatively, IFN may impact the immune infiltrate or, most likely, impact both the stroma and the infiltrate. However this is not addressed in the manuscript. This should be tested by deleting the IFNAR1 in the tumor cells combined with IFNAR $-/-$ mice or neutralization antibodies.

5- Cisplatin is not considered an immunogenic (i.e. IFN) inducing chemotherapeutic, which could be potentially explained by the results in the manuscript. However, another platinum-based chemotherapeutic, oxaliplatin, does induce IFN. Comparing the ability of the oxaliplatin and cisplatin to induce AhR driven feedback inhibition could reveal key information regarding induction of IFN and feedback inhibition in cancer cells.

Minor point

1- It would be nice to show altered STAT1 phosphorylation when the AhR-STING pathway is manipulated since STAT1 KO cells are used in the experimental approach.

Reviewer #6 (Remarks to the Author): with expertise in bladder cancer

Ma et al. report a large number of experiments covering a broad range of areas which are of interest not only to bladder cancer but also more generally. The work is generally well performed, although I have some concerns. A major, general problem, is that the concentrations of amino acids or other molecules used in the experiments are not fully relevant to the physiological/organismal conditions. This applies, for example, to the experiments in Figure 2 C: the serum concentration of tryptophan in humans and mice is in the range of 50-70 μM , which means that the effects tested largely fail to reproduce the physiological setting. Similarly, in Figure 3D, the concentrations of kynurenine used are

extremely high, non-physiological.

A number of statements in the text are biased and not substantiated by the data or current evidence. For example, the efficacy of the clinical trials of amino acid combination therapies is largely an overstatement. The authors should pay careful attention to this.

The paper needs English editing.

Specific comments

1. Please provide more information on the SYBC1 and MB49 cell lines. This is very important since one of the regions more commonly deleted in bladder cancer (in human tumors) is 9p21 where the IFN gene cluster sits.
2. It is not clear whether the experiments performed, and the results presented, correspond to technical or biological replicates. Of course, it should be the latter.
3. When immunofluorescence data are shown, please show a low magnification picture as well as an inset with a higher magnification.
4. The statistical tests applied are often inappropriate: for small sample size and non-normal distribution, Student' T test should NOT be applied.

Minor comments

1. The term "myometrial" used in the first paragraph of the introduction is misleading and should be replaced with "muscle".

Reviewer #1 (Remarks to the Author):

In conclusion, through the systematic experiment, they described how the Trp-IDO1-Kyn metabolic pathway influences IFN-I production. Eventually, they identified that Trp metabolism-AhR-STING pathway could affect the efficacy of the treatment. While the research is certainly important to understand the mechanism, the study was well conducted. Some comments are given below to enhance the manuscript further.

Response to R1:

Thank you for the positive comments and support for our work. We have followed up with more analyses as you recommended.

Page 14 of the manuscript (PDF version), “In addition, Trp deprivation or knocking-out IDO1 also decreased the ubiquitination of STING, while this could be reversed by adding Kyn (Figures 4E, S4D-E).” Readers could misunderstand this sentence. I recommend saying your result more explicitly.

Response:

We appreciate the reviewer's suggestion, and we have rewritten the description of these results and rearranged the panels in the revised manuscript. The new version of description has been added on page 16, line 13-16 of the revised manuscript: “in addition, knocking-out of IDO1 in SYBC1 cells resulted in decreased ubiquitination of STING under the treatment of cis-platin or cGAMP; however, the downregulation of STING ubiquitination led by knocking-out of IDO1 was reversed in presence of Kyn. Consistently, deprivation of tryptophan in the medium also resulted in decreased levels of STING ubiquitination, and addition of Kyn could also reverse this effect.”.

Page 17 of the manuscript, regarding molecular docking and MD simulations, the question is whether STING exists as a homodimer in the transmembrane domain of ER. However, the authors used the monomer of STING for molecular docking and MD simulation. The author could elaborate on how the monomer interacting with AhR in the study could elucidate the interaction between AhR and homodimer in nature. The question is whether the AhR binds the monomer differs from how the AhR binds the dimer.

Page 18 of the manuscript says, “Then, based on Molecular Mechanics Poisson-Boltzmann Surface Area (MMPBSA), we found that the bind free energy upon AhR with STING for AhR-STING and AhR-STING-Kyn was shifted from -127.36 kcal/mol to 171.43 kcal/mol.” It was a huge change in the binding energy between two systems “from -127.36 kcal/mol to 171.43 kcal/mol.” Please confirm this observation and

calculations. Right now, in AhR-STING-Kyn system, its binding energy is positive. So, the binding is not stable.

Also, ideally, more simulations should be performed to draw a more robust conclusion. I understand the difficulty of doing extra calculations; however, the authors must address this issue properly and justify their analysis.

Response:

The reviewer raises legitimate concerns from a perspective of molecular docking and MD simulation. First, we would like to explain the clerical error in the manuscript. Regarding the change of binding energy between AhR and STING monomer after binding to Kyn, we lost a "minus" when processing the manuscript. The value should indeed be -171.43 kcal/mol. The root means square deviation (RMSD) analysis showed that AhR and STING were indeed easier to form a stable system after binding with Kyn (original Figure 5D). Here, we have listed various components that constitute the total free binding energy (Table R1.1), hoping to have a clearer elucidation.

Table R1.1

The binding free energy (in kcal/mol) and its components obtained from the MM/PBSA calculation for AHR-STING and AHR-STING-KYN.

Contribution	AHR-STING	AHR-STING-KYN
ΔE_{vdw}	-228.84	-290.83
ΔE_{ele}	-521.89	-469.01
ΔG_{polar}	653.51	626.61
$\Delta G_{\text{nonpolar}}$	-30.15	-38.19
ΔG_{total}	-127.36	-171.43

Secondly, we have performed molecular docking and MD simulation on the interaction between AhR and STING dimer. We simulated two possible modes of interaction between AhR and the STING dimer: 1. One AhR molecule interacts with the C-terminus of the STING homodimer; 2. Two AhR molecules interact with the C-terminus of the STING dimer in mirror-image symmetry way. We simulated these two scenarios and analyzed together with previous data on the interaction of AhR with the STING monomer.

RMSD analysis indicated that either a single AhR molecule interacting with STING dimer (AhR-STING (monomer-dimer) or two AhR molecules interacting with STING dimer (AhR-STING (dimer) could reach a stable state very quickly within the simulation time; and both appeared to be more stable than the interaction of AhR with STING monomer (Figure R1.1).

Figure R1.1

Figure R1.1: The tendency of the root-mean-square-deviation (RMSD) plot for AHR-STING, AHR-STING (Dimer), and AHR-STING (Monomer-Dimer) complex models.

However, we found that in both cases involved STING dimer, a single molecule of AhR interacted with a single molecule of STING (Figure R1.2A-C); and the interacting amino acids with STING of all three scenarios were highly similar (Table R1.2-1.4), which were mainly concentrated in the C-terminus of AhR; this also indicated that the situation where one AhR molecule interacted with two STING molecules simultaneously did not exist. Moreover, in co-immunoprecipitation experiments with AhR truncation mutations in our previous manuscript (original Figure 5B), we also found that the N-terminus of AhR (1-424aa) hardly interacted with STING.

Table R1.2

The contact list between AHR with STING for AHR-STING.

Chain A	Residue	Chain B	Residue	Interaction type
AHR	Tyr371.O	STING	Gln227.NE2	Hydrogen bond
AHR	Asn373.ND2	STING	Thr229.OG1	Hydrogen bond
AHR	Asn373.ND2	STING	Asp237.O	Hydrogen bond
AHR	Gly374.N	STING	Gln227.OE1	Hydrogen bond
AHR	His625.ND1	STING	Asn188.ND2	Hydrogen bond
AHR	His626.O	STING	Arg191.NH1	Hydrogen bond
AHR	Lys628.NZ	STING	Glu249.OE1	Hydrogen bond, Salt bridge
AHR	Lys628.N	STING	Arg253.O	Hydrogen
AHR	Glu633.OE1/OE2	STING	Arg191.NH1/NH2	Hydrogen bond, Salt bridge
AHR	Glu633.OE1	STING	Arg253.NH1	Salt bridge
AHR	Gln640.OE1	STING	Asn187.N	Hydrogen bond
AHR	Gly649.O	STING	Gln227.NE2	Hydrogen bond
AHR	Met650.SD	STING	Gln227.NE2	Hydrogen bond
AHR	Gln658.OE1	STING	Lys224.NZ	Hydrogen bond
AHR	Leu815.N	STING	Asp210.O	Hydrogen bond
AHR	Asn816.O	STING	Gln266.NE2	Hydrogen bond
AHR	Asn817.ND2	STING	Thr263.OG1	Hydrogen bond
AHR	Asn817.ND2	STING	Gln266.OE1	Hydrogen bond
AHR	Asn820.O	STING	His232.NE2	Hydrogen bond
AHR	Leu827.O	STING	Arg94.NH1	Hydrogen bond
AHR	Pro829.CA	STING	Cys91.SG	Hydrogen bond
AHR	His831.N	STING	Arg86.O	Hydrogen bond
AHR	Glu835.OE1	STING	Arg76.NH1/NH2	Hydrogen bond, Salt bridge
AHR	Glu835.OE1	STING	Lys150.NZ	Hydrogen bond, Salt bridge

Table R1.3

The contact list between AHR with STING for AHR-STING (Dimer).

Chain A	Residue	Chain B	Residue	Interaction type
AHR	Tyr371.O	STING	Gln227.NE2	Hydrogen bond
AHR	Asn373.ND2	STING	Thr229.OG1	Hydrogen bond
AHR	Asn373.ND2	STING	Asp237.O	Hydrogen bond
AHR	Asn373.N	STING	Gln227.OE1	Hydrogen bond
AHR	Arg375.NH1/NH2	STING	Asn237.OD1/OD2	Hydrogen bond, Salt bridge
AHR	Ala606.O	STING	Arg191.NE	Hydrogen bond
AHR	His626.O	STING	Arg191.NH1	Hydrogen bond
AHR	Lys628.NZ	STING	Glu249.OE1/OE2	Hydrogen bond, Salt bridge
AHR	Gln629.OE1	STING	Arg191.NH1	Hydrogen bond
AHR	Glu633.OE1/OE2	STING	Arg191.N	Hydrogen bond
AHR	Glu633.OE1	STING	Arg253.NH1	Salt bridge
AHR	Glu633.O	STING	Arg253.NH1	Hydrogen bond
AHR	Gln640.NE2	STING	Leu189.O	Hydrogen bond
AHR	Gly649.O	STING	Gln227.NE2	Hydrogen bond
AHR	Asn655.OD1	STING	Lys224.NZ	Hydrogen bond
AHR	Gln658.OE1	STING	Lys224.NZ	Hydrogen bond
AHR	Pro812.O	STING	Asn211.ND2	Hydrogen bond
AHR	Asn816.ND2	STING	Asp210.O	Hydrogen bond
AHR	Asn816.O	STING	Gln266.NE2	Hydrogen bond
AHR	Asn817.OD1	STING	Thr263.OG1	Hydrogen bond
AHR	Asn817.ND2	STING	Thr263.OG1	Hydrogen bond
AHR	Asn817.ND2	STING	Gln266.OE1	Hydrogen bond
AHR	Thr821.OG1	STING	Gln266.OE1	Hydrogen bond
AHR	Pro833.O	STING	Gln276.NE2	Hydrogen bond
AHR	Ala836.O	STING	Gln276.NE2	Hydrogen bond
AHR	Arg837.NE	STING	Gln273.O	Hydrogen bond

Table R1.4

The contact list between AHR with STING for AHR-STING (Monomer-Dimer).

Chain A	Residue	Chain B	Residue	Interaction type
AHR	Tyr371.O	STING	Gln227.NE2	Hydrogen bond
AHR	Lys372.NZ	STING	Asp.237.OD2	Salt bridge
AHR	Asn373.ND2	STING	Asp237.O	Hydrogen bond
AHR	Asn373.N	STING	Gln227.OE1	Hydrogen bond
AHR	Arg375.NH1/NH2	STING	Asn237.OD1/OD2	Hydrogen bond, Salt bridge
AHR	Ser604.OG/N	STING	Asn188.OD1	Hydrogen bond
AHR	Ala606.O	STING	Arg191.NE	Hydrogen bond
AHR	Gln627.NE2	STING	Glu337.OE1	Hydrogen bond
AHR	Lys628.N	STING	Arg253.O	Hydrogen bond
AHR	Lys628.NZ	STING	Glu249.OE1/OE2	Hydrogen bond, Salt bridge
AHR	Gln629.O	STING	Gln252.NE2	Hydrogen bond
AHR	Gln629.OE1	STING	Arg191.NH1	Hydrogen bond
AHR	Glu633.OE1	STING	Gly192.N	Hydrogen bond
AHR	Glu633.OE1	STING	Arg253.NH1	Salt bridge
AHR	Gln640.NE2	STING	Leu189.O	Hydrogen bond
AHR	Gln658.OE1	STING	Lys224.NZ	Hydrogen bond
AHR	Pro812.O	STING	Asn211.ND2	Hydrogen bond
AHR	Asn816.ND2	STING	Asp210.O	Hydrogen bond
AHR	Asn817.ND2	STING	Thr263.OG1	Hydrogen bond
AHR	Thr821.OG1	STING	Gln266.OE1	Hydrogen bond
AHR	Ser834.O	STING	Gln276.NE2	Hydrogen bond

Furthermore, we calculated the binding free energy for a single AhR molecule and a single STING in three different cases, and found that the scenario of two AhR molecules and STING dimer interacted in mirror-symmetry pattern was the lowest (Table R1.5).

Table R1.5

The binding free energy (in kcal/mol) and its components obtained from the MM/PBSA calculation for AHR-STING, AHR-STING (Monomer-Dimer), and AHR-STING (Dimer).

Contribution	AHR-STING	AHR-STING (Monomer-Dimer)	AHR-STING (Dimer)
ΔE_{vdw}	-228.84	-236.65	-245.07
ΔE_{ele}	-521.89	-514.74	-504.57
ΔG_{polar}	653.51	646.08	641.32
$\Delta G_{nonpolar}$	-30.15	-33.15	-33.80
ΔG_{total}	-127.36	-138.45	-142.13

A study (PMID: 24001774) has shown that AhR could form homodimer through its own PAS domain (120-424aa). Therefore, we speculate that in the real world, the interaction mode of two molecules of AhR and STING dimer in mirror-symmetry pattern is more likely

to exist.

Thirdly, we have furtherly simulated the interaction between AhR and STING dimer in mirror-symmetry pattern after binding Kyn; and conducted further analysis with the new data and the previous data on the interaction between AhR and STING monomer in the original manuscript. In this scenario, we found that when AhR bind Kyn, both complexes took less time to reach steady state than the unbound state; however, comparing with interaction with STING monomer, the time to enter stable state was shorter in interaction of AhR with STING dimer (Figure R1.3).

Figure R1.3

Figure R1.3: The tendency of the root-mean-square-deviation (RMSD) plot for AHR-STING, AHR-STING-KYN, AHR-STING (Dimer), and AHR-STING-KYN (Dimer) complex models.

Moreover, root-mean-square-fluctuation (RMSF) analysis indicated that after binding Kyn, the fluctuation of STING was more stable in both models; while AhR (chain A) was more stable in the complex interacting with the STING dimer than with the STING monomer (Figure R1.4).

Figure R1.4

Figure R1.4: The root-mean-square-fluctuation (RMSF) plot for AHR residues and STING residues of AHR-STING, AHR-STING-KYN, AHR-STING (Dimer), and AHR-STING-KYN (Dimer) complex models.

Furthermore, the most stable binding modes between AHR and STING in the AHR-STING,

AHR-STING-KYN, AHR-STING (Dimer) and AHR-STING-KYN (Dimer) complex were shown (Figure R1.5A-D).

Figure R1.5

Figure R1.5: (A-D) The interaction between AHR and STING for AHR-STING (A), AHR-STING-KYN (B), AHR-STING (Dimer) (C), and AHR-STING-KYN (Dimer) (D). AHR is colored with marine, STING with magenta, KYN with yellow. The key residues in AHR are shown as cyan sticks while others as marine stick. The key residues in STING are shown as cyan sticks while others as magenta stick. The red dashes represent hydrogen bond interaction. The blue dashes represent salt bridge.

The contact lists between AHR and STING of these different scenarios were shown respectively (see above Table R1.2, R1.3 and Table R1.6 and R1.7 below).

Table R1.6

The contact list between AHR with STING for AHR-STING-KYN.

Chain A	Residue	Chain B	Residue	Interaction type
AHR	Gln627.OE1	STING	Arg253. NH1	Hydrogen bond
AHR	Gln629.OE1	STING	Arg253.N	Hydrogen bond
AHR	Gln637.NE2	STING	Asp223.OD2	Hydrogen bond
AHR	Lys643.NZ	STING	Asp223.OD1	Hydrogen bond, Salt bridge
AHR	Asn648.OD1	STING	Tyr186.OH	Hydrogen bond
AHR	Asn653.ND2	STING	Gln227.OE1	Hydrogen bond
AHR	Asn804.ND2	STING	Met214.SD	Hydrogen bond
AHR	Asn808.ND2	STING	Asp210.OD1	Hydrogen bond
AHR	Asn808.OD1	STING	Asn211.ND2	Hydrogen bond
AHR	Tyr811.O	STING	Asn211.ND2	Hydrogen bond
AHR	Glu814.N	STING	Asp210.O	Hydrogen bond
AHR	Glu814.OE1/OE2	STING	Lys224.NZ	Hydrogen bond, Salt bridge
AHR	Leu815.N	STING	Asp210.O	Hydrogen bond
AHR	Asn816.ND2	STING	Pro209.O	Hydrogen bond
AHR	Asn816.O	STING	Thr263.OG1	Hydrogen bond
AHR	Leu827.O	STING	Ser162.OG	Hydrogen bond
AHR	Arg837.NH2	STING	Asp301.OD1	Hydrogen bond, Salt bridge
AHR	Pro838.O	STING	Arg14.NH1	Hydrogen bond
AHR	Ser845.OG	STING	Arg14.NE	Hydrogen bond

Table R1.7

The contact list between AHR with STING for AHR-STING-KYN (Dimer).

Chain A	Residue	Chain B	Residue	Interaction type
AHR	Gln624.O	STING	Arg191.NH1	Hydrogen bond
AHR	Hic625.ND1	STING	Gly192.N	Hydrogen bond
AHR	Gln627.OE1	STING	Arg253. NH1	Hydrogen bond
AHR	Gln629.OE1	STING	Arg253.N	Hydrogen bond
AHR	Gln629.NE2	STING	Arg253.O	Hydrogen bond
AHR	Gln637.NE2	STING	Asp223.OD2	Hydrogen bond
AHR	Lys643.NZ	STING	Asp223.OD1	Hydrogen bond, Salt bridge
AHR	Asn648.O	STING	Gln227.NE2	Hydrogen bond
AHR	Asn653.ND2	STING	Gln227.OE1	Hydrogen bond
AHR	Lys801.NZ	STING	Gly207.O	Hydrogen bond
AHR	Asn804.ND2	STING	Met214.O	Hydrogen bond
AHR	Asn808.ND2	STING	Asp210.OD2	Hydrogen bond
AHR	Asn808.OD1	STING	Asn211.ND2	Hydrogen bond
AHR	Tyr811.O	STING	Asn211.ND2	Hydrogen bond
AHR	Glu814.N	STING	Asp210.O	Hydrogen bond
AHR	Glu814.OE1/OE2	STING	Lys224.NZ	Hydrogen bond, Salt bridge
AHR	Asn816.ND2	STING	Pro209.O	Hydrogen bond
AHR	Asn816.O	STING	Thr263.OG1	Hydrogen bond
AHR	Arg837.NH2	STING	Asp301.OD1	Hydrogen bond, Salt bridge
AHR	Pro838.O	STING	Arg14.NH1	Hydrogen bond

In general, after AhR bind to Kyn, the region where AhR interacted with STING was more concentrated at the C-terminus (all after the 600th amino acid). Then, we calculated the binding free energy for a single AhR molecule and a single STING in scenarios of binding Kyn, and found that Kyn led to decreased binding free energy, in both cases of interacting with STING monomer or STING dimer (Table R1.8). Moreover, the binding free energy decreased more in interaction with STING dimer after AhR bound Kyn than the interaction with STING monomer.

Table R1.8

The binding free energy (in kcal/mol) and its components obtained from the MM/PBSA calculation for AHR-STING, AHR-STING-KYN, AHR-STING (Dimer), and AHR-STING-KYN (Dimer).

Contribution	AHR-STING	AHR-STING-KYN	AHR-STING (Dimer)	AHR-STING-KYN (Dimer)
ΔE_{vdw}	-228.84	-290.83	-245.07	-337.58
ΔE_{ele}	-521.89	-469.01	-504.57	-446.10
ΔG_{polar}	653.51	626.61	641.32	609.93
$\Delta G_{nonpolar}$	-30.15	-38.19	-33.80	-40.92
ΔG_{total}	-127.36	-171.43	-142.13	-214.66

For the sake of showing the difference of binding mode between AHR and STING when the KYN binding to AHR-STING generally, the surface binding mode of AHR-STING and AHR-STING-KYN were shown. Obviously, the end of the residues in AHR are extended to STING, acting like a 'crab claw' chelating with STING (Figure R1.6 A, B), which resulted in stronger binding between AHR protein and STING protein. Importantly, above phenomenon was also confirmed in AHR-STING-KYN (Dimer) complex (Figure R1.6 C, D), which furtherly showed that no matter monomer or dimer that KYN could induce stronger binding between AHR and STING. For logical integrity, the relevant data on the interaction of AhR monomer with STING dimer was not included in the revised manuscript. The relevant data of the interaction between AhR monomer and STING monomer as well as AhR and STING dimer in mirror-symmetry pattern has been put in the revised manuscript. On page 19, line 13 to page 20, line 22.

Figure R1.6

Figure R1.6: (A-D) The surface binding model of AHR with STING for AHR-STING (A), AHR-STING-KYN (B), AHR-STING (Dimer) (C), and AHR-STING-KYN (Dimer) (D).

Pages 18 and 19, regarding ubiquitinations by K48 and K236. One part indicates that ubiquitinations reappeared when only K48 was reintroduced in the K0 mutation. Another part says, “the ubiquitination induced by AhR activation was restored after only R236 was reversed to lysine.” So, I could read that K48 is no longer necessary if AhR induces ubiquitination. What would be the insights from the mutation experiment for these two lysine residues?

Response:

The reviewer raises legitimate concerns on critical ubiquitination sites influenced by AhR; mainly related to the data in Figure 5 K-L and O-P of the original manuscript. Usually, ubiquitination modification on proteins exists in multiple forms of polyubiquitin chains. Polyubiquitin chains are usually formed through linking multiple ubiquitin molecules through their internal lysine (PMID: 33495455). Ubiquitin molecule contains 7 lysine: K6, K11, K27, K29, K33, K48, K63; among them, polyubiquitin chains linked by K27, K33, K48, or K63 are the most common (PMID: 33495455). Our previous data showed that AhR-induced STING undergone K48-linked polyubiquitin chain modification (Figure 5K-L in original manuscript); and the K236 site on STING was the place where the linkage happened (original Figure 5O-P). In another words, AhR-mediated ubiquitination, essentially is the linkage of K48-linked polyubiquitin chains to the K236 site of STING. Therefore, mutation of either K48 on ubiquitin or K236 on STING would abolish this AhR-mediated effect. We have revised the description of this part in the article to avoid misunderstandings.

Minor issues:

On Page 17, the manuscript says, “We found that the transcription activation domain-mediated the interaction with STING, rather than the PAS region that was previously reported to interact with other proteins (Figure 5B).” Please cite appropriate references

Response:

We appreciate the reviewer's kind reminder; we have cited the paper (PMID: 27721191, 28904176, 24001774) in the revised manuscript.

Figure S2 – Typo in the description

Response:

We appreciate the reviewer's kind reminder; we have made correction in the revised version.

Figure S5 (F) says, “Alignment of cGAS amino acid sequences.” This must be a typo. It should say, “Alignment of STING amino acid sequences.”

Response:

We appreciate the reviewer's kind reminder; we have made correction in the revised version.

Reviewer #3 (Remarks to the Author)

In this study, Ma et al. found that AhR acted as an adaptor that linked STING to Cul4B/RBX1 E3 ligase for proteasome degradation during cisplatin treatment in bladder cancer. Trp-IDO1-Kyn metabolic pathway regulated AhR-mediated STING ubiquitination and degradation. Pharmacologic inhibition of this pathway also increased STING stability and antitumor efficacy of cisplatin treatment in bladder cancer. Overall, this is an interesting study. Some concerns listed below need to be addressed properly.

Response to R3:

Thank you for the positive comments and support for our work. We have followed up with more analyses as you recommended.

1. Previous studies showed that STING activation-induced degradation was mainly controlled by lysosome, and proteasome played a minimal role during this process (PMID: 29241549). However, the current study showed that AhR-mediated proteasome degradation played a major role in controlling STING stability. To reconcile this discrepancy, the authors must demonstrate which pathway play a dominant role in regulating STING degradation during cisplatin treatment in the bladder cancer cells. This can be done by treating the cells with Cisplatin/cGAMP followed by MG132 or CQ treatment.

Response:

We are very grateful to the reviewer for a series of questions (Q1 and Q3) about the degradation mode of STING that brought a clearer understanding of the molecular mechanism underlying this work.

First, the biggest differences from previous studies (PMID: 29241549, 30842662) were that the treatments we used to activate STING did not induce significant degradation of STING itself. After exploration, we found it was mainly due to that under our experimental conditions, STING activation was weaker than those in previous studies. More importantly, we found that AhR in fact primarily degraded unactivated STING. The above conclusions came from the results of the following experiments we conducted:

1. According to the reviewer's suggestion, we used Cis-platin or cGAMP to treat SYBC1 cells and detected the STING protein levels in 12h and 24h in condition with or without MG132 or CQ pretreatment. In line with the previous data in our original manuscript (original Figure 2D, H, J), we found that treatment with Cis-platin or cGAMP did not lead to significant changes in STING protein levels in SYBC1, while STING protein levels were upregulated in the presence of MG132 or CQ (Figure R3.1A, B); and

compared to CQ, MG132 is more capable on upregulating STING protein.

Figure R3.1: Immunoassay for STING expression after indicated treatment.

2. Second, the degradation effect of lysosome on activated STING is particularly significant. The main reason is that the activated STING will eventually enter the endosome-lysosome pathways for degradation (PMID: 36918692). Here, we found that STING activation by chemotherapy, radiotherapy or even cGAMP was weak compared to conditions used in previous studies (Figure R3.2A). Moreover, we speculated that the ability of tumor cells to uptake cGAMP was also very weak, since we found that in presence of digitonin, an agent for solubilizing cellular membranes, protein levels of STING in SYBC1 cells showed dramatical decreasing when treated with cGAMP at same concentration as before (Figure R3.2B).

Figure R3.2: Immunoassay for pSTING (A) and STING (A, B) expression after indicated treatment.

In other words, in our scenario, due to the poor ability of the tumor cells to uptake cGAMP, the small amount of cGAMP entered showed similar effect of activating STING to chemoradiotherapy. Furthermore, under the treatment of cGAMP+digitonin, the degradation effect on STING of lysosome was indeed more significant than that of proteasome (Figure R3.3). Taken together, STING was in a weaker but persistent activation mode in our scenario.

Figure R3.3

Figure R3.3: Immunoassay for STING expression after indicated treatment.

3. Third, we further verified the degradation effect of AhR on unactivated STING. By using the endogenous AhR ligands Kyn, or the AhR exogenous ligand BAP, we found that in the absence of activation of STING (without chemo, RT or cGAMP), these compounds can also lead to a decrease in the protein level of STING, while MG132 but not CQ can reverse (Figure R3.4A). In line with this, our previous data also indicated that endogenous and exogenous AhR ligands could lead to increased ubiquitination levels on STING (original Figure 4J). Although the activation of STING was weak, the induced interferon can lead to a substantial upregulation of IDO1, since IDO1 is one of the genes most strongly affected by interferons (PMID: 23103127). Here, we further confirmed that IDO1 quickly plateaus, with no obvious dose effect with IFN- β (Figure R3.4B).

Figure R3.4

Figure R3.4: (A) Immunoassay for STING expression after indicated treatment; (B) mRNA expression of IDO1 after treated with IFN- β for 24h.

In conclusion, the damage of cellular DNA led by Cis-platin and radiotherapy is a continuous process, thus STING shows a weaker but persistent activation. Because of the weaker activation, less activated STING undergoes rapid degradation through lysosomes. On the other hand, the interferons led by the activation of STING can strongly up-regulate the expression of IDO1, resulting in production of Kyn. Kyn can lead to the degradation of the unactivated STING in the cell itself and the surrounding cells through AhR. The activation of STING is essentially the process of its aggregation in the endoplasmic reticulum to form oligomers and translocate (PMID: 31230712,

PMID: 32424334); and this AhR-mediated effect increases the threshold for aggregation and further STING activation.

It is also worth mentioning that, from the perspective of the cells, when STING is activated, there must be two states of STING in the cells, activated and unactivated. Our study essentially revealed an intrinsic inhibitory mechanism by which activated STING continuously suppressed the expression level of unactivated STING through downstream signaling. It is further suggested that the dependence of lysosome-dependent protein degradation and proteasome-dependent protein degradation may be different in regulating STING levels of different functional states. We have added and highlighted these relevant results in revised manuscript page 16, line 4-7 and page 16, line 22 to page 17, line 7.

2. In the Figure 1, does Trp affect IFN induction by other stimuli, such as DNA, RNA or LPS?

Response:

The reviewer raises legitimate concerns about the generality of our phenomenon. To this end, we transfected SYBC1 cells with HT-DNA, 3p-RNA or directly treated SYBC1 cells with LPS; after these treatments, cells were cultured in medium with or without Trp. After 24 hours, we detected the expression of IFNB1 via qPCR, and found that the omitting Trp had the most obvious effect on promoting IFNB1 expression in scenario of HT-DNA transfection. At the same time, removing Trp also enhanced the expression of IFNB1 induced by 3p-RNA but the enhancing effect was much weaker compared with the scenario of HT-DNA; while, omitting Trp showed no effect on the effect of LPS (Figure R3.5). We have added and highlighted these results in revised manuscript page 7, line 1-10.

Figure R3.5

Figure R3.5: mRNA expression of IFNB1 after transfection with HT-DNA, 3p-RNA or treated with LPS for 24h.

3. It has been known that STING undergoes lysosome-mediated degradation after

activation. However, in the Figure 2D and 2H, STING protein level was not decreased, rather it was increased by cisplatin or RT or cGAMP treatment. These data seemed to be in conflict with previous studies and should be verified. If so, can author comment on why STING was degraded after cGAMP treatment in those conditions?

Response:

We have addressed this question together in question 1.

4. In the Figure 3, was the antitumor effect of IDO1 inhibitor treatment dependent on STING? The 1-MT treatment should also be tested on STING KO cells. Similar for the Figure 4M and 4P, STING KO tumor model should also be tested with AhR inhibitors.

Response:

The reviewer raises legitimate concerns on whether the synergistic effect of IDO1 and AhR inhibitors (1-MT and Ch223191) with chemoradiotherapy depends on STING. To this end, we used WT and STING-KO MB49 cells, and by adjusting the number of injected cells, we ensured that there were no significant differences in tumor size between WT and STING-KO when the drug was administered. We found that knocking-out STING not only almost abolished the synergistic effect of AhR and IDO1 inhibitors but also decreased the efficacy of Cis-platin and RT (Figure R3.6A-D). We have added and highlighted these results in revised manuscript page 15, line 1-2 and page 19, line 8-10.

Figure R3.6

Figure R3.6: Effect of CH223191 and 1-MT combined with cisplatin or RT on MB49 growth (Mean \pm SEM).

5. In Figure 5, endogenous STING-AhR interaction should also be examined by immunoprecipitation. Whether their interaction was affected by Trp or kyn? K236 looked like the major ubiquitination site by AhR to induce STING degradation. However, all the experiments were performed with STING and AhR overexpression system. Whether K236 regulated STING degradation in the native condition should be examined. K236R CRISPR knock in cell line should be generated and tested for STING degradation and stability.

Response:

The reviewer raises legitimate concerns on endogenous interaction between AhR and STING and the effect of K236 in the native condition. First, for the interaction between AhR and STING under endogenous conditions, we have shown in the original data that the interaction between AhR and STING will become stronger in the presence of AhR ligands (original Figure 4H, J). We have emphasized this point in the revised version. Second, following the reviewer's suggestion, we constructed K236R knock-in SYBC1 cells through method based on Cas9 and oligonucleotide strands (PMID: 24157548). Mutation sites and sequencing data are shown here (Figure R3.7).

Figure R3.7: Sequencing of parental and individual clones of parental SYBC1 cells with knock-in expression of STING1(K236R) mutants. The red line indicates the sgRNA-targeting sequence. The black line indicates the PAM. Black arrows indicate mutated nucleotides. A mutated amino acid and its wild-type counterpart are indicated by the solid red box.

Consistent with previous data, endogenous K236R-STING had a longer degradation half-life under cGAMP treatment than wild-type (Figure R3.8A). At the same time, under the treatment of Kyn, the protein level of K236R-STING was also higher than that of the wild type and the ubiquitination level was lower in K236R-STING mutant (Figure R3.8B, C). More importantly, cells expressing endogenous K236R-STING showed higher expression of IFNB1 than WT cells under cisplatin or cGAMP treatment (Figure R3.8D). **We have added and highlighted these results in revised manuscript page 23, line 7-17.**

Figure R3.8

Figure R3.8: (A) Immunoassay for STING expression in K236R cells after indicated treatment in different time points; (B) Immunoassay for STING expression in K236R cells after indicated treatment. (C) Coimmunoprecipitation and immunoassay for extracts of cell lysate of K236R cells by anti-STING antibodies; (D) mRNA expression of IFNB1 of K236R cells after treated with cisplatin or cGAMP.

6. In the Figure 6, The E3 ligase Cullin 4B should have many other substrates that may also affect antitumor immune response. Whether the antitumor effect observed in Cullin 4B KO cells was dependent on STING? The authors should compare tumor growth between Cullin 4B KO and Cullin 4B/STING double knock out cells. Similar in the Figure 7, the control experiments were not well designed. The tumor growth should be monitored in parallel between SLC7A5 KO cells and SLC7A5/STING double KO cells. STING KO tumors should also be tested for the synergistic antitumor effect of Cis+BCH.

Response:

The reviewer raises legitimate concerns on whether the antitumor effect of Cullin 4B and SLC7A5 was dependent upon STING. To this end, we constructed Cullin4B/STING and SLC7A5/STING double knockout MB49 cell lines (Figure R3.9A). Interestingly, we found that both Cullin4B/STING DKO and SLC7A5 DKO cells formed bigger tumors than the WT cells (Figure R3.9B-C). This also indicated that Cullin4B and SLC7A5 were only part of the regulation of STING. Moreover, we also tested the effect of Cis+BCH on STING KO MB49 cells. In line with previous results, in STING-KO tumors, the treatment of Cis+BCH did not show obvious synergistic effect (Figure R3.9D). We have added and highlighted these results in revised manuscript page 25, line 1-3 and page 26, line 7-13.

Figure R3.9

Figure R3.9: (A) knocked-out efficiency verification by immunoassay; (B-C) Effect of knocked out indicated genes on MB49 growth (Mean \pm SEM); (D) Effect of BCH combined with cisplatin on MB49 growth (Mean \pm SEM)

Reviewer #4 (Remarks to the Author)

The study “AhR Diminishes the Efficacy of Chemotherapy via Suppressing STING Dependent Type-1 Interferon in Bladder Cancer” by Ma, Z. et al. identifies a novel mechanism by which AhR decreases STING levels through physical interaction with STING. The authors demonstrate that both genetic and pharmacological inhibition of AhR caused an increase in STING expression and correlative IFN-I levels. Furthermore, the study implicates this finding as an oncogenic mechanism through which bladder cancer (BC) cells can repress IFN-I levels which would otherwise stimulate antitumor immunity. This study is significant not only due to the characterization of a novel contributor to immune evasion, but due to the discovery of a physical connection between the kynurenine and STING pathways. However, additional work and clarifications are necessary to more thoroughly establish the major findings of the paper.

Response to R4:

Thank you for the positive comments and support for our work. We have followed up with more analyses as you recommended.

Major comments:

1. The authors indicate that, in BC, tumor cells are responsible for the majority of IFN-I_s in the tumor microenvironment. However, they only analyzed 4 out of 13 IFN-I_s. Given that myeloid-derived cells, particularly dendritic cells, are classically considered as primary contributors to IFN-I_s, authors should either clarify that tumor cells are responsible for the majority of those specific IFN-I subtypes or expand their study to include a majority of IFN-I_s.

Response:

The reviewer raises legitimate concerns on the source of IFN-I_s. We first put our conclusion here: Both tumor cells and myeloid immune cells are important sources of IFN-I_s in tumors, but at least for bladder cancer, tumor cells are the main source. We reached this conclusion through the analysis of the following aspects: First, we detected the expression of remaining subtypes of type I interferons that we did not test in the original manuscript (Figure R4.1). Like the previous results, the expression of these unchecked subtypes was also in comparable levels between tumor cells and myeloid cells from patients. We have added and highlighted these results in revised manuscript page 5, line 5-9.

Figure R4.1

Figure R4.1: mRNA expression of indicated IFN-Is in paired tumor and myeloid cells isolated from bladder cancer samples.

Second, single-cell sequencing data of bladder cancer and adjacent tissues from our previous study (PMID: 36459995) were analyzed; we found that most IFN-I subtypes could not be detected in sequencing which might be due to the depth of sequencing, while the epithelial cells expressed all the detected ones, and the expression levels were relatively higher (Figure R4.2A). On the other hand, we analyzed the ability of different types of cells to produce IFN-Is by three different gene sets related to IFN-Is production. In line with previous results, with single-cell sequence data, we found that tumor cells consistently showed the highest expression in all three different gene sets (Figure R4.2B).

Figure R4.2

Figure R4.2: (A) IFN-Is expression of each major cell type in scRNA-seq data; (B) Comparison of the type-I interferon production-related pathways between fibroblast cells, myeloid cells, and tumor cells in scRNA-seq data. P-value compared to fibroblast cells.

Furthermore, we analyzed the expression of IFN-I subtypes in other single-cell sequencing.

Macroscopically, tumor cells and myeloid immune cells were indeed the most likely to have certain levels of expression (Figure R4.3A). Third, studies (PMID: 30559422, PMID: 25344738) got a similar view to ours. Especially in this study (PMID: 30559422), the expression of downstream genes of interferon (ISGs) was used to illustrate the production of interferons. In their data, the purity of the tumor was not inversely proportional to the level of ISG (Figure R4.3B: original extended data Fig.2a in PMID: 30559422); more importantly, a large part of tumors with high levels of ISG also showed very low DC and NK/T cell content; and the level of ISG in bladder cancer cell lines was at a high level among all tumor cell lines (Figure R4.3C: original extended Data Fig.3b in PMID: 30559422). Fourth, the mechanisms we discovered here were applicable regardless of whether the source of IFN-Is was tumor or myeloid cells. **Considering the integrity of the logic, the above data was not included in the revised manuscript.**

Figure R4.3

Figure R4.3: (A) mRNA expression of IFN-Is in public scRNA-seq database; (B) Correlation between ISG core score and tumor purity in TCGA primary tumors; (C) ISG core score in PDX and CCLE samples.

2. Furthermore, regarding the BC patients the cells were isolated from, there is no mention of what therapeutic interventions, if any, the patients were undergoing as that may contribute to the status of IFN-1 concentrations.

Response:

The reviewer raises legitimate concerns on whether the expression of IFN-Is in patients was interfered by ongoing therapeutic interventions. The patient samples we tested for interferons expression in original manuscript (original Figure S1F, G) were obtained at the

time of diagnosis, before the patients started treatment. We have added additional information on the revised figure legend.

3. In Figure 1, the authors demonstrate that Trp-deficiency along with cis-plat treatment increased CD8+ T cell infiltration. However, the experiment was relatively short at 3 weeks. While the field has moved away from the thought that Kyn-AhR tolerization is driven through Trp deprivation, it is unclear if Trp deprivation would not become a concern in Trp-starved patients. Therefore, authors should address this in their Discussion.

Response:

We appreciate the reviewer's suggestion, and we have included this in the discussion; highlighted in page 29, line 16-22.

4. Can Kyn decrease STING at lower concentrations? 400uM Kyn is well beyond the amount necessary to promote AHR translocation, which following the proposed model, should be the driver decreasing STING. STING decrease should be visible between 10-40uM of Kyn. 400uM Kyn would be cytotoxic to most cells. Therefore, authors must repeat experiments using lower amounts of Kyn and verify that the same effects are still observed for all findings. Effects of Kyn in cell growth, proliferation, Ahr translocation and target gene activation also must be verified.

Response:

The reviewer raises legitimate concerns on whether Kyn could still exert obvious effect at a lower concentration. To this end, we reformulated Kyn's stock solution and aliquoted it in the smallest possible volume to ensure that each aliquoted Kyn was only used once. We started from 10uM and tested the effect on SYBC1 cells of 6 concentration gradients; respectively 10uM, 20uM, 40uM, 80uM, 160uM, 320uM. After Kyn treatment, we detected cell proliferation, apoptosis, AhR activation and STING protein levels. We found that starting from 160uM, Kyn showed cell growth inhibition effect (Figure R4.4A); but even at 320uM, SYBC1 cells did not show obvious apoptosis (Figure R4.4B).

Figure R4.4

Figure R4.4: (A) 2×10^5 cells were seed; cell number were detected after treated with Kyn for 24h; (B) The percentage of apoptotic cells (Annexin V+/7AAD-, Annexin V-/7AAD+, and Annexin V+/7AAD+) after treated with Kyn for 24h.

From 20uM, AhR began to show obvious signs of activation; but when the concentration of Kyn was greater than 80uM, further increase in AhR activation brought by additionally increased concentration of Kyn was not obvious (Figure R4.5A, B).

Figure R4.5

Figure R4.5: (A) Immunoassay for AhR expression in the nucleus of cells after treated with Kyn; (B) mRNA expression of CYP1A1 after treated with Kyn.

From 20uM, the protein level of STING began to decrease, and in the range of 20uM-160uM, the concentration effect of Kyn was more obvious (Figure R4.6). Considering that the concentration range of Kyn in tumors is 10-80uM, we retested the Kyn-related experiments in our manuscript at the concentration of 40uM (original Figure 2D-F; 3C-E, H; 6A-D, F; supplementary figure 3F-G; 4C-F, I; 5D-E). We added these data in the revised manuscript and replaced the original data with new data.

Figure R4.6
SYBC1

Figure R4.6: (A) Immunoassay for STING expression in cells after treated with Kyn.

5. Given that IDO1 is implicated as responsible for the activation of AhR along with the antitumoral phenotype being CD8+ T cell driven, it is surprising that the authors do not measure or mention IFN-g. While an IFN-II, IFN-g is the best characterized inducer of IDO1 and is an inflammatory cytokine secreted by CD8+ T cells. How does cis-plat induction of IDO1 compare to IFN-g? Do IFN-g levels change similarly with IFN-Is?

Response:

The reviewer raises legitimate concerns on the role of IFN- γ in our scenario. IFN- γ is mainly produced and secreted by T cells, NKT cells and NK cells (PMID: 17063185); while IFN-Is could be secreted by all nucleated cells. We examined the expression of IDO1 in SYBC1 after Cis-platin or IFN- γ (10ng/ml) treatment, and found that IFN- γ induced more potent IDO1 expression compared with Cis-platin (Figure R4.7A). Regarding the changes of IFN- γ levels, we found that neither chemotherapy nor radiotherapy could induce the expression of IFN- γ in tumor cells under the conditions of in vitro experiments (Figure R4.7B). But for the in vivo model of MB49, Cis-platin treatment increased the content of IFN- γ in the tumor (Figure R4.7C), which may be related to the increased infiltration of CD8+ T cells (original Figure 1L, O). For integrity of the logic, this part of the data on IFN- γ was not included in the revised manuscript. Considering the integrity of the logic, the above data was not included in the revised manuscript.

Figure R4.7

Figure R4.7: (A) Immunoassay for IDO1 expression after indicated treatment; (B-C) mRNA expression of IFNG for indicated cells (B) or MB49 bearing mice (C) after indicated treatment. P-value compared to DMSO.

[RESPONSE AND FIGURE REDACTED]

Other comments:

- 1. The ending of the abstract should be re-written for clarity.**

Response:

We appreciate the reviewer's suggestion, and we have re-written this part in the revised

manuscript.

2. “Adoptive immunity” presumably should be adaptive immunity.

Response:

We are sorry for the typo, and we have corrected in the revised manuscript.

3. Authors need to clarify that IDO1 is not the sole rate-limiting enzyme.

Response:

We appreciate the reviewer's suggestion, and we have changed the description in the revised manuscript.

Reviewer #5 (Remarks to the Author)

The report is an interesting and novel exploration of the role for IFN-driven feedback that may promote immune suppression following treatment with platinum-based chemotherapy. The main strength of the report is the novel link identified between non canonical AhR function and reduction of STING signaling that may reduce IFN production and immune-mediated reduction of tumor burden in bladder cancer. Overall the experiments are well designed and thorough, describing a function interaction that results in ubiquitin-mediated reduction of STING and attenuation of STING signaling. However there are some significant points that somewhat diminish enthusiasm for what is otherwise an interesting report.

Response to R5:

Thank you for the positive comments and support for our work. We have followed up with more analyses as you recommended.

Major points-

1- The manuscript relies entirely on in vivo results with one bladder cancer cell line, MB49. The authors should show the applicability across multiple models of bladder cancer using spontaneous models, if possible.

Response:

The reviewer raised legitimate requests for more validation of our findings in this manuscript in multiple animal models. Spontaneous models of bladder cancer are mainly induced by chemicals (**PMID: 29367767**), especially N-butyl-N-(4-hydroxybutyl)-nitrosamine (BBN); more importantly, compared to spontaneous animal models induced by gene mutations in other cancers (breast cancer, lung cancer), the success rate of compound-induced bladder cancer is relatively low, usually only 60-80%; and we have no relevant experience which makes success rate even lower. On the other hand, the time for successful model construction usually takes more than six months; and in order to obtain more reliable data, this process will be longer. In consideration of timeliness, please forgive us that we really do not have the ability to construct spontaneous models of bladder cancer and conduct related experiments.

However, consider the importance of validating our findings in multiple animal models. We further constructed MB49 cell line based orthotopic bladder cancer model and verified our findings. Firstly, by sgGFP-MB49 and sgSTING-MB49 cells, we found that, consistent with previous data, deprivation of Trp could sensitize the efficacy of cis-platin, whereas knocking-down of STING nearly abolished this sensitization effect (**Figure R5.1A**).

Moreover, we also verified the effects of various inhibitors tested in manuscript with orthotopic bladder cancer model, and reached consistent conclusions (Figure R5.1B). We added these data in the revised manuscript in page 12, line 5-12 and page 25, line 12-16.

Figure R5.1: (A) Effect of Trp free F with cisplatin on orthotopic MB49 growth (Mean \pm SEM); (B) Effect of BCH, 1-MT, and CH223191 combined with cisplatin on orthotopic MB49 growth (Mean \pm SEM).

2- In figure 1k mice were placed on Trp-free chow for 21 days. It is surprising that mice could tolerate the loss of Trp for this extended amount of time as extended loss of Trp can cause cellular/tissue pathology, behavioral/cognitive issues, and eventual mortality. It would be appropriate to address this caveat in the description of the experiment.

Response:

The reviewer raised legitimate concerns for tolerance towards Trp-deficient diet in mice. In fact, we tested the tolerance of mice to Trp deprivation prior to the formal experiments; and we found no deaths (0/8) even up to receiving 30 days' Trp-deficient diet and there was no obvious decline in vitality and abnormal behavior; however, the mice began to lose hair around the 10th day, and the body weight decreased ~5% around the 20th day. Furthermore, we also detected the Trp levels in plasma of mice with HPLC-MS and found

that the Trp levels decreased to 40% after receiving Trp-deficient diet for 3 days compared with those received normal diet; decreased to 20% on the sixth day and further decreased to 10% on the tenth day (Figure R5.2). We added these data in the revised manuscript in page 7, line 12-14.

Figure R5.2: Tryptophan levels in mice serum detected by HPLC-MS after Trp free F. P-value compared to 0d.

On the other hand, for the subcutaneous tumor model of MB49, after inoculation of 1×10^6 cells, tumors usually grew to $5 \times 5 \text{mm}^3$ to $7 \times 7 \text{mm}^3$ on the 7-10 day and could be evenly grouped according to tumor sizes. Thus, further operations usually started from the seventh day. Trp deprivation began on day 7, while cis-platin treatment was initiated two days after Trp deprivation. Ultimately, the Trp deprivation period was 14 days. Since the side effects caused by the deprivation of Trp within 14 days were not that obvious, our related experiments were all conducted under this condition. In the revised manuscript, we have described the experimental procedures in more detail, and add related description of the side effects caused by Trp deprivation. Specifically, the description was added in page 7, line 15 in the revised manuscript.

3- Despite the fact that the main hypothesis proposes that AhR limits immune-responses post therapy, there a complete lack of immune characterization of the tumors +/- Trp, +/- AhR inhibition, or +/- IDO inhibition. This should be done do demonstrate that alterations in the immune infiltrate composition or activation state accompany alterations to the aforementioned pathways to further strengthen the argument.

Response:

The reviewer raised legitimate request for more confirmation of alterations in immune environment led by inhibition Trp-IDO-AhR pathway. In the manuscript, we mainly found that Trp-IDO-AhR pathway affected the production of IFN-Is, and IFN-I signaling played an

important role in the infiltration and activation of T cells (PMID: 26027717). Therefore, we focused on analyzing the effects of Trp deprivation, AhR or IDO1 inhibition on T cell infiltration and function during cis-platin treatment. Consistent with previous results, we found that either Trp deprivation, inhibition of AhR or inhibition of IDO1 increased T cell infiltration (Figure R5.3A-D). On the other hand, we also found that the expression of IL2 and IFN γ in T cells also increased after inhibiting this pathway; while compared with deprivation of Trp and inhibition of IDO1, the increase of IFN γ and IL2 in T cells after inhibiting AhR was more obvious (Figure R5.3A-D). This may be related to the fact that AhR also induced CD8+ T cell exhaustion (PMID: 29533786, PMID: 33432230). We added these data in the revised manuscript in page 7, line22 to page 8, line 2; page 15, line 3-6; and page 19, line 10-12.

Figure R5.3

Figure R5.3: (A) flow cytometry plot of gating strategy; (B-D) percentage of CD3, 8⁺ cells; IL2, CD3, 8⁺ cells; and IFNG, CD3, 8⁺ cells in MB49 bearing mice after treated with Trp free F (B), 1-MT (C), and CH223191 (D) combined with cisplatin.

4- IFN may impact the tumors directly altering immunogenicity, MHC expression, etc.. Alternatively, IFN may impact the immune infiltrate or, most likely, impact both the stroma and the infiltrate. However this is not addressed in the manuscript. This should be tested by deleting the IFNAR1 in the tumor cells combined with IFNAR^{-/-} mice or neutralization antibodies.

Response:

The reviewer raised legitimate request for further validation of effects of IFN-Is. Thus, according to the suggestion of the reviewer, we generated IFNAR1 knockout MB49 cell line (Figure R5.4A). Then, we constructed tumor models by subcutaneous injection of sgGFP-MB49 cells or sgIFNAR1-MB49 cells. We found that sgIFNAR1-MB49 cells formed larger tumors than sgGFP-MB49 cells when inoculated with the same number of cells (Figure R5.4B). We furtherly increased the number of inoculated sgGFP-MB49 cells to ensure consistent tumor sizes upon cisplatin treatment. After cis-platin treatment, we found that tumors formed by sgIFNAR1-MB49 cells were also larger, and the difference was bigger than the scenario without Cis-platin treatment (Figure R5.4C).

Figure R5.4: (A) knocked-out efficiency verification by immunoassay; (B) Effect of knocked out IFNAR1 on MB49 growth (Mean \pm SEM); (C) Effect of knocked out IFNAR1 combined with cisplatin on MB49 growth (Mean \pm SEM).

Furthermore, we collected tumors two days after the first dose of Cis-platin and detected the content of intratumoral IFN- β , the expression of key ISGs that related to adaptive immunity (H2-kb, Cxcl10) and IFN-Is (IFNB1, IFNA4). The changes of intratumoral T cells were also tested via flow cytometry four days after the first dose of Cis-platin. We found that after knocking out IFNAR1, there was no significant change in intratumoral IFN- β levels and the expression of IFNB1 and IFNA4; while the expression of these key ISGs (H2-kb, Cxcl10) and the T-cell infiltration were decreased (Figure R5.5A-C).

Figure R5.5: (A) IFN- β content in MB49 bearing mice after indicated treatment; (B) mRNA expression of IFNB1, IFNA4, H2-Kb, and CXCL10 in MB49 bearing mice; (C) percentage of CD8⁺ T cells in tumor.

Then, we further explored whether IFN-Is affected the efficacy of Cis-platin through other non-tumor cells by using IFNAR1 blocking antibody. In the absence of cisplatin treatment, sgIFNAR1-MB49 cells formed tumors of almost the same sizes regardless of injection of IFNAR1 neutralizing antibody (Figure R5.6A). Under Cis-platin treatment, intraperitoneal injection of IFNAR1 neutralizing antibody in sg-IFNAR1 MB49 tumor bearing mice led to larger but not statistically different tumor volumes (Figure R5.6A). In line with this, there were no differences in the expression of key ISGs in the presence of IFNAR1 neutralizing antibody, but the T cell infiltration showed decreased trend (Figure R5.6B).

Figure R5.6

Figure R5.6: (A) Effect of indicated treatment on MB49 growth (Mean \pm SEM); (B) mRNA expression of H2-Kb, and CXCL10 and percentage of CD8⁺ T cells in tumor in MB49 bearing mice.

Finally, we further examined whether Cis-platin combined with Trp deprivation, IDO1 inhibitor or AhR inhibitor still had synergistic effect in sgIFNAR1-MB49-formed tumors. As expected, knocking-out IFNAR1 almost erased the synergistic effect of depriving Trp or inhibiting IDO1, while inhibiting AhR showed dampened but existed synergistic effect (Figure R5.7). We added these IFN-I signaling related data in the revised manuscript in page 8, line 3 to page 14, line 15. For logical completeness, the data related with combination therapy was not included in the revised manuscript.

Figure R5.7

Figure R5.7: (A-C) Effect of knocked out IFNAR1 combined with Trp free F (A), 1-MT (B), and CH223191 (C) MB49 growth (Mean \pm SEM).

5- Cisplatin is not considered an immunogenic (i.e. IFN) inducing chemotherapeutic, which could be potentially explained by the results in the manuscript. However, another platinum-based chemotherapeutic, oxaliplatin, does induce IFN. Comparing the ability of the oxaliplatin and cisplatin to induce AhR driven feedback inhibition could reveal key information regarding induction of IFN and feedback inhibition in cancer cells.

Response:

The reviewer raised legitimate concerns on immune activation mediated by Cis-platin through IFN-Is and suggested to use oxaliplatin as a comparator. First, the reason why we chose Cis-platin was that most neoadjuvant chemotherapy (NAC) for bladder cancer used chemotherapy regimens containing Cis-platin in clinical practice. Second, the therapeutic efficacy of Cis-platin also depended on the regulation of the immune system (PMID: 25204552), and the combined use of Cis-platin and immune checkpoints also showed a synergistic effect (PMID: 28592566,36100320); moreover, such immune regulation modes also included activation of cGAS-STING dependent IFN-Is production (PMID: 28279982, 36306685, 30518877, 32678307).

Then, we detected the regulatory effects of Cis-platin and oxaliplatin on the activation of STING and production of IFN-Is on SYBC1 and UMUC-3 cells at two time points (24h, 48h) and three concentration gradients (2uM, 20uM, 80uM). However, we found that at 48h, tumor cells treated with 20uM and 80uM Cis-platin were most dead and could not be used to collect protein samples. Thus, we only analyzed samples treated for 24 hours. We found that Cis-platin induced stronger STING phosphorylation and higher IFNB1 expression at 2uM and 20uM; but at 80uM, activation of STING by Cis-platin was not further enhanced; while oxaliplatin at 80uM induced more potent activation effects than Cis-platin of all the three concentrations (Figure R5.8A, B).

Figure R5.8

Figure R5.8: (A) Immunoassay for STING expression after indicated treatment; (B) mRNA expression of IFNB1 after indicated treatment.

Consistent with previous results, IDO1 expression, AhR activation, and Kyn production were directly proportional to the amount of IFN-Is expression, that 20uM cis-platin was more effective than 20uM oxaliplatin, while 80uM oxaliplatin was the most effective in our experiments (Figure R5.9A-C).

Figure R5.9: (A) mRNA expression of IDO1 and CYP1A1 after indicated treatment; (B) Intracellular Kyn concentration detected by HPLC-MS after indicated treatment.

Moreover, deprivation of Trp and inhibition of AhR or IDO1 also enhanced the effects of oxaliplatin on STING activation and IFN-Is induction (Figure R5.10A-C). For logical completeness, the relevant data comparing the activation of STING and IDO1-AHR pathway by Cis-platin and oxaliplatin were not included in the revised manuscript. We added the relevant data on inhibiting AhR driven feedback loop leading to enhanced STING activation by oxaliplatin into the revised manuscript. The content was on page 11, line 1-3; page 14, line 7-9; and page 18, line 16-19.

Figure R5.8: (A) Immunoassay for pSTING/STING and pSTAT1 expression after treated with Ox combined with Trp free M (A), 1-MT (B), and CH223191 (C);

Minor point

1- It would be nice to show altered STAT1 phosphorylation when the AhR-STING pathway is manipulated since STAT1 KO cells are used in the experimental approach.

Response:

The reviewer raised legitimate request on providing more data on STAT1 phosphorylation.

In our revised manuscript, we added the data of STAT1 phosphorylation in Figure 2-3 when the IDO1-AHR pathway was manipulated. Specifically, these new data was added in our revised Fig. 2E, 2F, 2I, S2E; Fig. 3E, 3F, S3I.

Reviewer #6 (Remarks to the Author):

Ma et al. report a large number of experiments covering a broad range of areas which are of interest not only to bladder cancer but also more generally. The work is generally well performed, although I have some concerns. A major, general problem, is that the concentrations of amino acids or other molecules used in the experiments are not fully relevant to the physiological/organismal conditions. This applies, for example, to the experiments in Figure 2 C: the serum concentration of tryptophan in humans and mice is in the range of 50-70 microM, which means that the effects tested largely fail to reproduce the physiological setting. Similarly, in Figure 3D, the concentrations of kynurenine used are extremely high, non-physiological.

A number of statements in the text are biased and not substantiated by the data or current evidence. For example, the efficacy of the clinical trials of amino acid combination therapies is largely an overstatement. The authors should pay careful attention to this.

The paper needs English editing.

Response to R6:

Thank you for the positive comments and support for our work. We have followed up with more analyses, brief here:

1. We have re-conducted extensive analysis to further confirm that lower concentration of Trp still worked in our system.
2. We also re-examined the lowest dose of Kyn that could be effective, and re-performed related experiments.
3. We tuned-down some description in the revised manuscript, especially the part about amino acid metabolism related combination therapy in the introduction and discussion.
4. We have conducted language editing in revised manuscript and hope that our revised writing will be easier for reviewer and readers to read.

The data related with the verification of the effective concentrations of Trp and Kyn in vitro experiments are listed below:

1. Data related with Trp:

Typically, the concentration of Trp in RPMI 1640 medium is 25uM. Our original data have shown that this concentration already exhibited a significant effect compared to 1640 medium without Trp (original Figure 1F, H, supplementary figure 1F). In order to avoid the batch effect caused by the degradation of Trp due to storage, we re-formulated Trp stock

solution and aliquoted it in the smallest possible volume to ensure that each aliquoted Trp was only used once. At the same time, each batch of formulated Trp stock solution was only used for two weeks. Under this condition, we retried four concentrations of Trp at 25uM, 50uM, 100uM and 200uM. Consistent with the previous results, under Cis-platin treatment, the protein levels of STING also decreased significantly and showed a gradual decline trend with the increase of Trp concentration (Figure R6.1). Furthermore, we reconducted experiments related with supplementing Trp in original Figure 1-2 and supplementary figure1-2; the experiments were conducted with the concentration of Trp at 50uM. These data were added in revised Fig. 2C, 2E, S2C, S2D, S2E.

Figure R6.1: Immunoassay for STING expression after indicated treatment.

2. Data related with Kyn:

Reviewer 4 raised similar concerns about Kyn:

“4. Can Kyn decrease STING at lower concentrations? 400uM Kyn is well beyond the amount necessary to promote AHR translocation, which following the proposed model, should be the driver decreasing STING. STING decrease should be visible between 10-40uM of Kyn. 400uM Kyn would be cytotoxic to most cells. Therefore, authors must repeat experiments using lower amounts of Kyn and verify that the same effects are still observed for all findings. Effects of Kyn in cell growth, proliferation, Ahr translocation and target gene activation also must be verified.”

Thus, we list the responses to Reviewer 4 below:

Response:

To this end, we reformulated Kyn's stock solution and aliquoted it in the smallest possible volume to ensure that each aliquoted Kyn was only used once. We started from 10uM and tested the effect on SYBC1 cells of 6 concentration gradients; respectively 10uM, 20uM, 40uM, 80uM, 160uM, 320uM. After Kyn treatment, we detected cell proliferation, apoptosis, AhR activation and STING protein levels. We found that starting from 160uM, Kyn showed cell growth inhibition effect (Figure R4.4A); but even at 320uM, SYBC1 cells did not show obvious apoptosis (Figure R4.4B).

Figure R4.4

Figure R4.4: (A) 2×10^5 cells were seeded; cell number were detected by flow cytometry after treated with Kyn for 24h; (B) The percentage of apoptotic cells (Annexin V+/7AAD-, Annexin V-/7AAD+, and Annexin V+/7AAD+) after treated with Kyn for 24h.

From 20uM, AhR began to show obvious signs of activation; but when the concentration of Kyn was greater than 80uM, further increase in AhR activation brought by additionally increased concentration of Kyn was not obvious (Figure R4.5A, B).

Figure R4.5

Figure R4.5: (A) Immunoassay for AhR expression in the nucleus of cells after treated with Kyn; (B) mRNA expression of CYP1A1 after treated with Kyn.

From 20uM, the protein level of STING began to decrease, and in the range of 20uM-160uM, the concentration effect of Kyn was more obvious (Figure R4.6). Considering that the concentration range of Kyn in tumors is 10-80uM, we retested the Kyn-related experiments in our manuscript at the concentration of 40uM (original Figure 2D-F; 3C-E, H; 6A-D, F; supplementary figure 3F-G; 4C-F, I; 5D-E). We added these data in the revised manuscript and replaced the original data with new data.

Figure R4.6

Figure R4.6: (A) Immunoassay for STING expression in cells after treated with Kyn.

1. Please provide more information on the SYBC1 and MB49 cell lines. This is very important since one of the regions more commonly deleted in bladder cancer (in human tumors) is 9p21 where the IFN gene cluster sits.

Response:

The reviewer raised legitimate request on providing more data on SYBC1 and MB49 cell lines to exclude the interference caused by the mutation of IFN-Is related genes. To this end, we performed whole-genome sequencing on SYBC1 and MB49 cells, and found that in these two cell lines, the genes encoding IFN-Is did not appear frameshift, fusion and other mutations that lead to the loss function of transcript proteins. We put the mutation-related data of these two cell lines in the attached excel files. The data of the sequencing matrix could be downloaded through to this link: (<https://pan.baidu.com/s/1UG750VvmHaRxoJsWYyiPWQ>. Extraction code: xs65)

2. It is not clear whether the experiments performed, and the results presented, correspond to technical or biological replicates. Of course, it should be the latter.

Response:

The data presented in the manuscript were biological replicates, not technical replicates. We have further noted this in the revised manuscript.

3. When immunofluorescence data are shown, please show a low magnification picture as well as an inset with a higher magnification.

Response:

We've added related pictures according to your suggestion. Please see Fig. 4J, 6G in the revised manuscript for details.

4. The statistical tests applied are often inappropriate: for small sample size and non-normal distribution, Student' T test should NOT be applied.

Response:

Thank you for your correction, we have replaced the original statistical method with either Wilcoxon test or two-tail t-test depending on the sample distribution and sample size, and these change does not affect our previous conclusions. Please see figure legend in the revised manuscript for details.

Minor comments

1. The term "myometrial" used in the first paragraph of the introduction is

misleading and should be replaced with "muscle".

Response:

We've replaced related description according to your suggestion.

REVIEWERS' COMMENTS

Reviewer #1 (Remarks to the Author):

A comprehensive revision of the original manuscript was done to reflect some concerns from the previous reviewers. In conclusion, through the systematic experiment, they described how the Trp-IDO1-Kyn metabolic pathway influences IFN-I production. Eventually, they identified that the Trp metabolism-AhR-STING pathway could affect the efficacy of the treatment. This study is significant due to the characterization of a novel contributor to immune evasion and the discovery of a physical connection between the Kyn and STING pathways.

Based on the authors' responses and results from new additional experiment works, this revised manuscript further enhances their findings and hypotheses regarding the Trp metabolism-AhR-STING pathway. This is worth publishing in the journal.

Reviewer #3 (Remarks to the Author):

The authors properly addressed my concerns.

Reviewer #4 (Remarks to the Author):

The manuscript was significantly improved by addition of new experiments and clarifications.

Reviewer #5 (Remarks to the Author):

The authors have addressed my concerns. I have no further substantive issues with the manuscript.

Reviewer #6 (Remarks to the Author):

The authors have adequately responded to my concerns.

Reviewer #1 (Remarks to the Author):

A comprehensive revision of the original manuscript was done to reflect some concerns from the previous reviewers. In conclusion, through the systematic experiment, they described how the Trp-IDO1-Kyn metabolic pathway influences IFN-I production. Eventually, they identified that the Trp metabolism-AhR-STING pathway could affect the efficacy of the treatment. This study is significant due to the characterization of a novel contributor to immune evasion and the discovery of a physical connection between the Kyn and STING pathways.

Based on the authors' responses and results from new additional experiment works, this revised manuscript further enhances their findings and hypotheses regarding the Trp metabolism-AhR-STING pathway. This is worth publishing in the journal.

Response to R1:

Thank you again for the positive comments and support for our work.

Reviewer #3 (Remarks to the Author):

The authors properly addressed my concerns.

Response to R3:

Thank you again for the positive comments and support for our work.

Reviewer #4 (Remarks to the Author):

The manuscript was significantly improved by addition of new experiments and clarifications.

Response to R4:

Thank you again for the positive comments and support for our work.

Reviewer #5 (Remarks to the Author):

The authors have addressed my concerns. I have no further substantive issues with the manuscript.

Response to R5:

Thank you again for the positive comments and support for our work.

Reviewer #6 (Remarks to the Author):

The authors have adequately responded to my concerns.

Response to R6:

Thank you again for the positive comments and support for our work.